# COP-Q: Safety-First Reinforcement Learning with Cholesky Ordered Projection

## Abstract

Using uncertainty in Q-values to mitigate overestimation, enhance exploration, and ensure safety has proven effective in single-objective deep Q-learning. However, when learning vector-valued Q-functions for correlated goals, uncertainties become intertwined across objectives. Conventional approaches either treat uncertainty in each objective independently or collapse them into one dimension, often resulting in unstable learning, low sample efficiency, limited exploration, and particularly unsafe behaviours. To address these challenges, this study proposes Cholesky Ordered Projection Q-learning (COP-Q), a novel method that guides safety-first exploitation and exploration using full multi-objective uncertainty. We first propose generalized multi-objective confidence bounds via covariance matrix factorization. For priority-ordered objectives, such as in safety-critical or cost-constrained reinforcement learning, Cholesky factorization is employed to incorporate inter-objective covariance into confidence bounds in a conditionally sequential manner. The lower bound yields conservative temporal difference targets to reduce overestimation, while the upper bound assigns optimistic Q-values to promote exploration. COP-Q is evaluated on standard MuJoCo and velocity-constrained SafetyVelocity-v1 benchmarks, demonstrating robust safety performance and competitive total returns. The proposed method is compatible with various deep Q-learning frameworks with minimal computational overhead, making it practical for a wide range of multi-objective and constrained reinforcement learning tasks.

## 1 Introduction

Many real-world control tasks involve multiple correlated objectives. When deploying Reinforcement Learning (RL) agents in such tasks, accurately estimating the expected return of each objective is essential for making nuanced, risk-aware decisions. Multi-Objective RL (MORL), for instance, estimates per-objective returns and aggregates them using scalarization—via weighted sums or nonlinear functions—to derive policies along the Pareto front (Hayes et al., 2022). Similarly, constrained RL separately estimates cumulative reward and cost to ensure satisfying safety or resource constraints (Wachi & Sui, 2020). Even using a fixed scalarization, safety-critical tasks require precise estimation of risk-related returns (Hoel et al., 2023). Therefore, learning vector-valued returns is fundamental for handling multiple objectives in RL.

Deep Q-learning, an off-policy RL framework known for high sample efficiency via experience replay, suffers from two key challenges: overestimation of Q-values (Thrun & Schwartz, 2014) and under-exploration (Ladosz et al., 2022). Quantifying Q-value uncertainty helps mitigate both. Overestimation stems from the Bellman update's max operator, which amplifies positive bias and destabilizes learning. A common solution is to use multiple independent Q-networks and derive a *conservative estimate* from their outputs (Van Hasselt et al., 2016). For exploration, Q-value uncertainty can serve as a bonus, forming an *optimistic estimate* that promotes visiting less-explored state-action pairs (O'Donoghue et al., 2018). Such uncertainty-driven strategies have been widely studied in single-objective Q-learning (Lockwood & Si, 2022).

However, when learning vector-valued Q-functions for correlated objectives, overestimation and exploration become entangled across objectives. Conventional methods either quantify uncertainty only in the scalarized total Q-value, as in MORL (Van Moffaert et al., 2013), or treat each objective independently, as in cost-constrained RL (Xu et al., 2022). While straightforward, both approaches

overlook inter-objective correlations, potentially causing the following consequences. **(1) Unstable learning:** In scalarization-based methods, two estimates with similar total Q-values may be distant in Q-space (Figure 1-(a)). This discrepancy allows estimation noise to dominate target selection in temporal difference (TD) updates, leading to unstable learning. **(2) Low sample efficiency:** When objectives are treated independently and per-objective lower bounds are combined, the resulting scalarized Q-value may become over-conservative, as the red point in Figure 1-(a), thus slowing down the learning. **(3) Under-exploration:** Collapsing inter-objective covariance into a scalar can obscure high uncertainty in individual objectives (Figure 1-(b)), leading the agent to overlook under-explored state-action pairs. **(4) Unsafe behaviours:** Conservatism in the total objective does not guarantee conservative estimates for safety (Figure 1-(b)). Overestimating the safety objective may hinder the safety learning and cause unsafe actions.

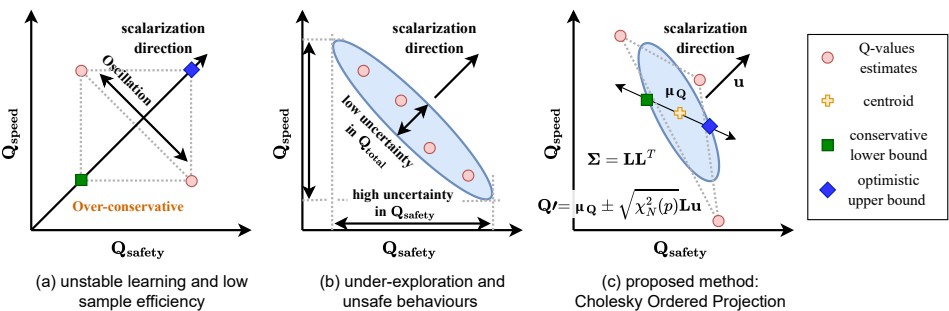

Figure 1: (a) and (b): Potential consequences of scalarization-based and independent approaches. (c): The proposed Cholesky Ordered Projection Q-learning (COP-Q). It smoothens the confidence bound estimation by an ellipsoid and guides prioritized exploitation-exploration by Cholesky factorization.

All these challenges highlight one fundamental knowledge gap:

*How can multi-objective uncertainty in Q-values guide exploitation and exploration in safety-critical deep Q-learning, thereby improving learning robustness, safety compliance, and sample efficiency?*

Central to this research question, we propose a novel method, **Cholesky Ordered Projection Q-learning (COP-Q)**. We first introduce generalized multi-objective confidence bounds for Q-values under linear scalarization. Geometrically, Q-value estimates are projected onto an iso-confidence ellipsoid in Q-space, as shown in Figure 1-(c). For safety-critical or constrained RL tasks with priority-ordered objectives, Cholesky factorization sequentially encodes covariance into the bounds, guiding safety-prioritized exploitation and exploration.

## 2 RELATED WORK

This study addresses the overestimation and exploration issues by leveraging uncertainty in vector-valued Q-function estimation. Relevant prior work is briefly overviewed below.

The max operator in vanilla Q-learning amplifies noisy value estimates, producing over-optimistic TD targets and unstable learning (Thrun & Schwartz, 2014). To mitigate this, Double-Q learning (Hasselt, 2010) decouples action selection and evaluation by maintaining two separate value functions, reducing overestimation. Clipped Double Q-learning (Fujimoto et al., 2018) further addresses this overestimation bias by taking the minimum of two independent target networks. Ensemble-based methods, such as Randomized Ensembled Deep Q-learning (REDQ) (Chen et al., 2021), MaxMin Q-learning (Lan et al., 2020), and SUNRISE (Lee et al., 2021), extend this idea to more than two Q-networks, using a high update-to-data ratio to achieve low bias and high sample efficiency. In the context of MORL, scalarization techniques are commonly used to convert vector-valued q-functions into a scalar, including linear (Van Moffaert et al., 2013) and non-linear formulations (Agarwal et al., 2022). Then, the single-objective remedies can be applied, for example, in the CAPQL (Lu et al., 2023). This is the so-called Scalarization of Expected Return (SER). Another approach is Expectation of Scalarized Return (ESR), which scalarizes the reward vector first and only learns the scalar total Q-function (Roijers et al., 2018). ESR is not relevant to this study, because it converts the problem into a single-objective task that is not aware of the risk. We do not introduce more.

Applying objective-wise double-Q or clipped minima is a second straightforward baseline (Abels et al., 2019). For example, in constrained Q-learning using PID Lagrangian multipliers (Stooke et al., 2020), the performance and cost Q-functions are learned independently. Preference-driven MORL (Basaklar et al., 2022) also uses per-objective double-Q targets to mitigate overestimation, but adds a cosine-similarity term to avoid collapsing to the objective-wise minimum. However, the methods above overlook the correlations between objectives.

For uncertainty-driven exploration, uncertainty-driven approaches estimate an optimistic upper confidence bound (UCB) of the Q-value. Classic methods include bandit-style UCB (Auer, 2002), Bootstrapped DQN (Osband et al., 2016), and OAC for continuous control (Ciosek et al., 2019). SUNRISE (Lee et al., 2021) shows that ensembles can yield high-quality UCBs in both discrete and continuous settings. In MORL, Pareto Q-learning (Van Moffaert & Nowé, 2014) maintains vector Q-values and uses set evaluation to explore the Pareto front. Recent approaches also explore scalarization vector spaces, such as UCB-guided utility search (Shi et al., 2024) and Envelope Q-learning (Yang et al., 2019), which samples diverse scalarization weights to generalize across preferences. However, most of these works focus on scalarization-based exploration, leaving open the core question: how to leverage uncertainty in vector-valued Q-functions to guide exploration under arbitrary scalarization weights.

In summary, to the best of our knowledge, few studies address the overestimation and exploration issues by using uncertainty in vector-valued Q-functions. This paper fills this gap by incorporating inter-objective covariance into Q-learning frameworks, particularly for safety-critical RL tasks.

## 3 PROBLEM FORMULATION

We consider a Markov decision process defined by $(S, A, R, p, p_0, \gamma)$. The state space is $S \subseteq \mathbb{R}^m$. For an observed state $s_t \in S$, an agent takes action $a_t$ in the action space $A \subseteq \mathbb{R}^d$, the environment transitions to the next state with a probability distribution $p(s_{t+1}|s_t, a_t)$, and the agent receives a vector reward $\mathbf{r}_t \in \mathbb{R}^N$, defining $N$ objectives. The tuple $(s_t, a_t, r_t, s_{t+1})$ is a transition stored in a replay buffer. The initial state distribution is $p_0(s_0)$. The agent is controlled by a policy $a \sim \pi(\cdot|s)$. The action-value function (Q-function) of each objective is defined by:

$$Q_i^\pi(s, a) = \mathbb{E}_\pi \left[ \sum_{t=0}^\infty \gamma r_i(s_t, a_t) \mid s_0 = s, a_0 = a \right], \tag{1}$$

where $\gamma \in (0, 1)$ is the shared discounting factor. The multi-objective vector-valued Q-function is noted as $\mathbf{Q}^\pi(s, a) \in \mathbb{R}^N$. A linear scalarization function converts $\mathbf{Q}^\pi(s, a)$ to a total objective $Q_{\text{total}}^\pi$:

$$Q_{\text{total}}^\pi(s, a) = \mathbf{u}^\intercal \mathbf{Q}^\pi(s, a), \qquad \text{with} \qquad \|\mathbf{u}\|_2 = 1. \tag{2}$$

We aim to learn the vector-valued Q-function and a policy that maximizes the expected total return for a given or changing scalarization weight $\mathbf{u}$. This setting is fundamental. For example, MORL learns a set of policies corresponding to different fixed scalarization functions, and constrained RL often adapts $\mathbf{u}$ dynamically using Lagrangian multipliers (Ray et al., 2019).

This study focuses on two key components of Q-learning. (1) how to compute the next Q-vector $\mathbf{Q}'(s_{t+1}, a')$ in the temporal difference (TD) target $\mathbf{y}_t$:

$$\mathbf{y}_t = \mathbf{r}_t + \gamma \mathbf{Q}'(s_{t+1}, a'), \quad a' = \arg\max_a \mathbf{u}^\intercal \mathbf{Q}'(s_{t+1}, a), \tag{3}$$

And (2) how to compute an uncertainty-based exploration bonus to form an optimistic upper bound of the vector-values Q estimates.

## 4 METHOD: CHOLESKY ORDERED PROJECTION Q-LEARNING (COP-Q)

This section first presents the theoretical basis for the proposed multi-objective confidence bounds, then details the use of Cholesky factorization for safety-critical or objective-prioritized tasks, followed by the practical implementation of COP-Q.

### 4.1 GENERAL FORM OF MULTI-OBJECTIVE CONFIDENCE BOUNDS

We first generalize the confidence bounds of the Q-value from a scalar to a vector. In single-objective Q-learning, assume that the Q-value (expectation of return) is a random variable following a Gaussian $p(Q) = \mathcal{N}(Q|\mu_Q, \sigma_Q^2)$ (D'Eramo et al., 2021). For simplicity, we omit $(s, a)$ and superscript $\pi$. A *confidence level* $p \in (0, 1)$ determines a confidence region bounded by two end points:

$$\mu_Q \pm \sqrt{\chi_1^2(p)}\sigma_Q, \tag{4}$$

enclosing probability $p$. Here $\chi_1^2(p)$ is the quantile of the Chi-squared distribution. When estimating vectorized multi-objective Q-values, under the Gaussian assumption $p(\mathbf{Q}) = \mathcal{N}(\mathbf{Q}|\boldsymbol{\mu_Q}, \boldsymbol{\Sigma_Q})$, the confidence level $p$ now determines an *ellipsoid* confidence region centred at the mean:

$$(\mathbf{Q} - \boldsymbol{\mu_Q})^\intercal \boldsymbol{\Sigma_Q}^{-1} (\mathbf{Q} - \boldsymbol{\mu_Q}) = \chi_N^2(p), \tag{5}$$

where $\mathbf{Q} = [Q_1, \cdots, Q_N]^\intercal$, and $\chi_N^2(p)$ is the squared Mahalanobis distance. Now the confidence bound is not a unique point, we therefore construct a projected point on this ellipsoid considering the scalarization weight $u$. A natural choice is applying a linear transformation:

$$\mathbf{Q_p} = \boldsymbol{\mu_Q} + \mathbf{Au}. \tag{6}$$

Replacing $\mathbf{Q}$ in equation 5 by $\mathbf{Q_p}$, we seek the general form of $\mathbf{A}$, which admits infinite solutions. If $\boldsymbol{\Sigma_Q}$ is factorized into "squared-root forms" with positive entries:

$$\boldsymbol{\Sigma_Q} = \mathbf{CC}^\intercal \;\Rightarrow\; (\mathbf{Au})^\intercal (\mathbf{C}^\intercal)^{-1} \mathbf{C}^{-1}(\mathbf{Au}) = \chi_N^2(p) \;\Rightarrow\; ||\mathbf{C}^{-1}\mathbf{Au}|| = \sqrt{\chi_N^2(p)}, \;\forall\, \mathbf{u}, \tag{7}$$

then the solution is:

$$\mathbf{C}^{-1}\mathbf{A} = \pm\sqrt{\chi_N^2(p)}\mathbf{R} \;\Rightarrow\; \mathbf{A} = \pm\sqrt{\chi_N^2(p)} \times \mathbf{CR} \text{ s.t. } \mathbf{R}^\intercal \mathbf{R} = \mathbf{I}, \tag{8}$$

where $\mathbf{R}$ is an isometry, representing rigid-body rotation. The final projected points are:

$$\mathbf{Q_p} = \boldsymbol{\mu_Q} \pm \sqrt{\chi_N^2(p)} \times (\mathbf{CR})\mathbf{u}. \tag{9}$$

This family of solutions is informative because $\mathbf{C}^{-1}$ "whitens" the Gaussian to $\mathcal{N}(\boldsymbol{\mu_Q}, \chi_N^2(p)\mathbf{I})$. Conversely, $\mathbf{C}$ re-injects full inter-objective covariances into confidence bounds. To ensure that the projection aligns with the total objective, the condition $\mathbf{u}^\intercal (\mathbf{CR})\mathbf{u} \geq 0$ must hold. Otherwise, the scalarized lower bound could exceed the mean value, rendering it meaningless. In this sense, the rotation matrix $\mathbf{R}$ serves as a generalized "clipping" operator, extending the conventional single-objective clipped double-Q learning (Fujimoto et al., 2018) to the multi-objective setting. How to compute $\mathbf{R}$ is detailed in the Appendix A. For simplicity, from now on, we note $\mathbf{C}_{\text{clip}} = \mathbf{CR}$.

This raises the question of how to choose an appropriate matrix factorization. Since reward signals are typically unstructured vectors, incorporating task-specific structure can improve learning. For a wide range of tasks explicitly involving safety, we propose using Cholesky factorization.

### 4.2 CHOLESKY FACTORIZATION FOR PRIORITIZED OBJECTIVES

Assume that objectives are prioritized from highest to lowest, with safety or the constrained cost often coming first. For example, safety violations, such as a robot falling or a vehicle crashing, terminate the episode immediately. So, $p(\mathbf{Q})$ is decomposed by the chain rule:

$$p(\mathbf{Q}) = \prod_{i=1}^{N} p(Q_i|\mathbf{Q}_{1:i}), \tag{10}$$

which means that $\mathbf{C}$ must be *lower triangular*. The unique factorization is *Cholesky factorization* $\boldsymbol{\Sigma_Q} = \mathbf{LL}^T$. $\mathbf{L}$ encodes full multi-objective uncertainty following the priority structure of objectives. Specifically, each objective's confidence bound is conditioned only on higher-priority objectives, with the highest-priority objective relying solely on its marginal distribution. This *Cholesky Ordered Projection* (COP) prioritizes reducing overestimation and encouraging exploration in an ordered manner, while utilizing correlations to correct the excessive conservatism of independent approaches. Therefore, COP inherently guides ordered, prioritized exploitation and exploration.

The equation 9 are solutions of the whitening family only. *Eigen-decomposition* provides another pair of solutions in this category. Other non-whitening solutions include *extremum projection* and *diagonal projection*. We describe these alternatives in the Appendix B, and will compare them in our benchmark and ablation studies.

### 4.3 PRACTICAL IMPLEMENTATION OF COP IN DEEP Q-LEARNING

The implementation of COP follows the principle of minimality. For $N$ objectives, at least $N + 1$ estimates $\{\hat{\mathbf{Q}}^{(1)}, \cdots, \hat{\mathbf{Q}}^{(N+1)}\}$ are needed to compute the covariance matrix. As the sample size is close to dimensionality, *biased* covariance matrix estimator (denominator is $N + 1$) is used. For lower bound computation, we fix the Mahalanobis distance $\chi_N^2(p) = 1$:

$$\mathbf{Q}' = \hat{\boldsymbol{\mu}}_{\boldsymbol{Q}} - \mathbf{L}'_{\text{clip}}\mathbf{u}. \tag{11}$$

where $\hat{\boldsymbol{\mu}}_{\boldsymbol{Q}}$ denotes the estimated mean, and $\hat{\mathbf{L}}_{\text{clip}}$ is the clipped Cholesky factorization of the covariance matrix, i.e., **LR**. Notably, when $N = 1$, equation 11 is exactly clipped double-Q learning (Fujimoto et al., 2018). The biased estimator gives the half distance between two Q-values. So, the lower bound becomes the smaller Q-value. For upper bound computation, the hyperparameter $\chi_N^2(p_e)$ determines the level of optimism when facing uncertainty in exploration. A higher $\chi_N^2(p_e)$ means being more optimistic, saying, take a higher risk to pursue a potentially higher return.

$$\hat{\mathbf{Q}}_{\text{UB}} = \hat{\boldsymbol{\mu}}_{\boldsymbol{Q}} + \sqrt{\chi_N^2(p_e)} \times \hat{\mathbf{L}}_{\text{clip}}\mathbf{u}, \tag{12}$$

The COP-Q implementation is summarized as follows (add superscript $\pi$ for actor-critic methods):

1. **Initialization:** Rank $N$ objectives based on their priorities; use $N + 1$ independent Q-networks; each Q-network estimates the vector-valued Q-function; choose an exploration step length $\beta = \sqrt{\chi_N^2(p_e)}$ as the only hyperparameter.

2. **Training:** Estimate the biased covariance matrix of next Q-values, do Cholesky factorization and clip it to get $\mathbf{L}'_{\text{clip}}(s_{t+1}, a')$. The TD target for updating Q-networks is:

$$\mathbf{y}_t(s_{t+1}, a') = \mathbf{r}_t + \gamma[\hat{\boldsymbol{\mu}}_{\boldsymbol{Q}}(s_{t+1}, a') - \mathbf{L}'_{\text{clip}}(s_{t+1}, a')\mathbf{u}]. \tag{13}$$

   If using actor-critic methods, compute the actor update objective using the same formula:

$$Q^{\pi}_{\text{total}}(s_t, \pi(s_t)) = \mathbf{u}^{\mathsf{T}}[\hat{\boldsymbol{\mu}}_{\boldsymbol{Q}}(s_t, \pi(s_t)) - \hat{\mathbf{L}}_{\text{clip}}(s_t, \pi(s_t))\mathbf{u}]. \tag{14}$$

   where $\hat{\mathbf{L}}_{\text{clip}}$ is the clipped Cholesky factorization of the covariance matrix estimated from the critics' outputs using a biased estimator.

3. **Active exploration (optional)**: Compute the upper bound of total Q-value to guide action sampling in data collection:

$$\hat{Q}_{\text{UB}}(s, a) = \mathbf{u}^{\mathsf{T}}\hat{\boldsymbol{\mu}}_{\boldsymbol{Q}}(s, a) + \beta \times \mathbf{u}^{\mathsf{T}}\hat{\mathbf{L}}_{\text{clip}}(s, a)\mathbf{u}, \tag{15}$$

   where the last term is the multi-objective exploration bonus using COP.

This implementation introduces minimal computation overhead by using the minimal number of required critics and the most efficient factorization. Since TD target computation and exploration are essential to deep Q-learning, COP is compatible with most frameworks. The projected lower bound and the clipping rotation are visualized in the right part of Figure 2. In this example, COP prioritizes conservative learning for the firstly ordered objective (usually safety) while allowing more optimistic learning for the potentially conflicting objective of speed, thereby promoting safe exploration. Compared to clipped double-Q approaches, the ellipsoid smoothens the TD target, avoiding oscillations caused by the next Q-value selection. In terms of active exploration, COP can be combined with UCB (Auer, 2002) or Optimistic Actor-Critic (Ciosek et al., 2019), directly using equation 15 as the upper bound of Q-value.

## 5 EXPERIMENTS

We evaluate COP-Q on two sets of robot locomotion tasks using Brax (Freeman et al., 2021): standard MuJoCo environments and the SafetyVelocity-v1 benchmark (Ji et al., 2023). For the standard MuJoCo, we select five robots (see Table 1). The objective is to maximize a scalarized return while estimating each objective separately. Note that halfcheetah has no safety objective. It is selected as a reference case without clear prioritization between objectives. To amplify the effects of correlated and conflicting objectives, we increase control cost weights compared to the default

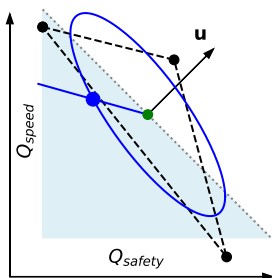 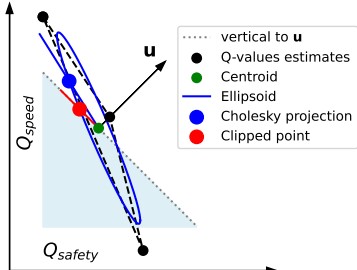

Figure 2: The projected lower bound point given by safety-first Cholesky factorization. In the first case, the projected point is in the lower-bound semi-plane. In the second case, the projected point is clipped by applying the rotation matrix $\mathbf{R}$, see Appendix A.

settings. SafetyVelocity-v1 (Zhang et al., 2020) is also based on MuJoCo. We select four robots that share the same reward structure. The goal is to maximize the scalarized total reward (using default weights) while keeping the agent's velocity return below a predefined safety threshold, same as the formulation in CAL (Wu et al., 2024) and CVPO (Liu et al., 2022). The two objectives are reward (total return) and cost (velocity return). Note that the "safety" here is defined by velocity rather than the safety objective itself. As the reward also contains the velocity term, the two objectives are systematically correlated, making the feasible set tight. This benchmark follows the Constrained MDP (CMDP) formulation (Altman, 2021). When using a primal-dual approach, the Lagrangian multiplier is updated during training, thus composes a changing scalarization weight $u$. We therefore use the latest in-place $u$ in each update iteration during training. Additional details are provided in Appendix C.

The standard MuJoCo benchmark includes two baselines: (1) *Scalarization double-Q* uses two independent critics; each estimates the vector Q-values. The critic with the lower scalarized total Q-value is used in TD targets and actor updates. (2) *Independent double-Q* uses $N$ pairs of critics to learn each objective independently. We also consider a reference method based on the proposed ellipsoid confidence bound for ablation: (3) *Extremum projection* uses the same number of critics as COP-Q but considers uncertainty along the scalarization direction only (see

Table 1: Standard MuJoCo environment settings. Unless noted, objectives are safety(+), velocity(+), control cost(-).

| Robot | Objectives | Scalarization weight |
|---|---|---|
| hopper | s, v, c | (1, 1, 1) |
| walker2d | s, v, c | (1, 1, 0.5) |
| humanoid | s, v, c | (1, 5, 1) |
| ant | s, v, c | (1, 1, 1) |
| halfcheetah | v, c (no s) | (1, 1) |

Table B.1). COP-Q and all baselines are implemented within the TD3 framework (Fujimoto et al., 2018), with constant action noise to ensure that performance differences stem from TD target and exploration strategies. Additionally, Fast Adaptive Multi-task Optimization (FAMO) (Liu et al., 2023) is applied to critic updates in all methods, preventing one objective from dominating the loss gradient. Further details are provided in Appendix D.

For SafetyVelocity-v1, we evaluate both on-policy and off-policy constrained RL baselines. On-policy methods include *CUP* (Yang et al., 2022), *RCPO* (Tessler et al., 2018), *PPOSaute* (Sootla et al., 2022), and *CPPOPID* (Stooke et al., 2020). Other recent on-policy baselines, such as RLSF (Reddy Chirra et al., 2024), are also promising candidates for future comparison on this benchmark. Off-policy baselines include *SACLag-UCB* (Stooke et al., 2020), which improves SACLag via conservative cost learning, and *CAL* (Wu et al., 2024), which combines conservative cost estimation with the augmented Lagrangian method (Luenberger et al., 1984). For fairness, we use 2 reward critics, 4 cost critics, and UTD=1 in CAL, omitting randomized ensembles. Both off-policy baselines learns two objectives independently. Our COP-Q is implemented on top of CAL (SAC-based) but replaces the critic update with our proposed COP mechanism, putting the cost objective in the first place and using only 3 critics to estimate the reward-cost covariance matrix. All other components remain identical to CAL. All methods are trained using 5 random seeds, similar to the experiments in CAL (Wu et al., 2024).

## 5.1 STANDARD MUJOCO

The experimental results of the standard MuJoCo are organized into two parts. First, we benchmark the overall learning performance of different TD target computation methods, without uncertainty-driven exploration. Next, we evaluate the effectiveness of the proposed COP active exploration strategy and its impact on sample efficiency.

**Main results** Figure 3 shows the mean and standard deviation of total and safety return training curves over five random seeds for all tasks. Table 2 shows the quantitative total returns. More statistics are provided in the Appendix E. Overall, COP-Q consistently outperforms or is on par with baselines in total return, with the most significant improvement observed on humanoid. For single-legged (hopper) and bipedal (walker2d, humanoid) robots that are prone to falling, COP-Q quickly attains the highest safety return (5000 for humanoid, 1000 for the others) and remains stable. This rapid learning of safety is due to the safety-first principle induced by Cholesky factorization. The scalarization double-Q exhibits oscillations in hopper and walker2d, reflecting instability in TD target selection. The independent double-Q performs well on humanoid and halfcheetah but fails on walker2d and ant, and converges slowly on hopper, due to its overly conservative TD target. Extremum projection achieves final total returns comparable to scalarization double-Q but with smoother learning curves, benefiting from our proposed ellipsoidal bounds that smooth TD target estimation. This observation also confirms that the advantage of COP-Q originates from its TD target computation, instead of using more critics. In halfcheetah, all methods have close performances, likely because the absence of a falling penalty decorrelates velocity and control cost, simplifying the multi-objective landscape.

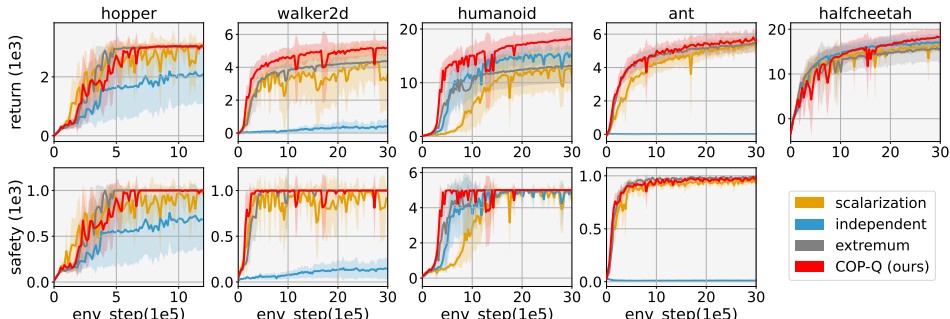

Figure 3: Training curves of COP-Q and baselines. Shaded areas mark standard deviations.

Table 2: Total return of the final policy after training (mean $\pm$ std, over 5 runs)

| Methods | Hopper | Walker2d | Humanoid | Ant | HalfCheetah |
|---|---|---|---|---|---|
| Scalarization | $3020 \pm 233$ | $3626 \pm 1402$ | $11283 \pm 5024$ | $5329 \pm 503$ | $16579 \pm 1369$ |
| Independent | $2203 \pm 924$ | $382 \pm 391$ | $14861 \pm 1468$ | $14 \pm 3$ | $17158 \pm 1946$ |
| Extremum | $2435 \pm 1190$ | $4369 \pm 420$ | $13137 \pm 3059$ | $5446 \pm 533$ | $15569 \pm 2846$ |
| **COP-Q** | $\mathbf{3042 \pm 99}$ | $\mathbf{5185 \pm 465}$ | $\mathbf{18209 \pm 1705}$ | $\mathbf{5573 \pm 498}$ | $\mathbf{18367 \pm 1863}$ |

**Objective ordering** The default objective order for COP-Q is safety–velocity–control (svc). If putting other objectives ahead of safety, such as control-velocity-safety (cvs) or velocity-safety-control (vsc), the top row of Figure 4 shows that the learning performance can be significantly degraded in some tasks. These results show that COP-Q is sensitive to the ordering of objectives. For robot locomotion, a reasonable ordering is prioritizing safety first, then improving velocity, and finally governing energy efficiency. This choice consistently gains high total returns in all five tasks.

**Structure of projection transformation** The bottom row of Figure 4 compares the projection methods listed in Table B.1, with extremum projection already presented in Figure 3. Among these methods, only Cholesky factorization consistently achieves high total returns and rapid convergence across all tasks. This performance gap is caused by different assumptions about the objective structure. The other projections use symmetric transformation matrices, which correspond to undirected graphs and therefore capture only correlations. In contrast, Cholesky factorization produces a lower triangular

matrix that defines ordered priority, incorporating the covariance matrix of Q-values into confidence bounds through a predefined directed structure. This inductive bias allows COP-Q to more effectively utilize inter-objective relationships, particularly in safety-critical tasks.

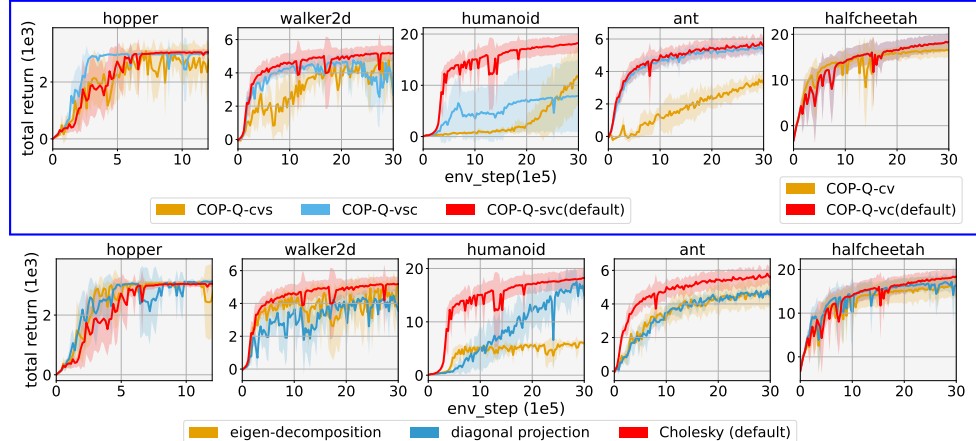

Figure 4: *Top row*: Training curves of COP-Q under different objective orders. Note that halfcheetah does not have a safety objective. *Bottom row*: Comparison of projection and covariance matrix factorization methods (details in the Appendix B).

**Exploration effectiveness**  To investigate whether COP can guide better exploration, we implement the upper confidence bound given by COP into the Optimistic Actor-Critic (OAC) exploration framework (Ciosek et al., 2019), and further combine it with REDQ (Chen et al., 2021), a well-known algorithm that uses an ensemble of critics and a high update-to-data (UTD) ratio to reduce estimation bias and improve sample efficiency. REDQ's large number of critics also increases the precision of covariance matrix estimation. Following REDQ's recommended settings, we use 10 critics, sample 2 random critics for each critic update, set the UTD ratio and actor delay to 20, and reduce the replay buffer size to 1/10 of the main benchmark. By doing so, the effect of uncertainty-driven exploration is isolated. We evaluate the extremum-OAC (considering uncertainty along the scalarization direction alone on the proposed ellipsoid confidence bound) and COP-OAC against baselines.

As shown in Figure 5, REDQ already achieves high sample efficiency (note environment steps are also 1/10 of the main benchmark), but COP-OAC converges faster than the REDQ baseline and REDQ + Extremum-OAC in 3 out of 4 environments, evidencing that guiding exploration by COP based on full multi-objective uncertainty is more effective than relying on scalarized uncertainty only. The variance of return remains high for *humanoid*. Using more random seeds cannot narrow the variance. For some random seeds, the policy is always stuck in suboptimal points. However, this high variance is not reported in the original single-objective REDQ (Chen et al., 2021). This high variance is likely due to the combination of high UTD and multi-objective learning. More investigations are needed to further clarify the reason. But for the non-stuck cases, COP-OAC still performs the best. We keep these original results for clarity.

In summary, COP-Q achieves robust learning, fast safety convergence, and competitive returns in the standard MuJoCo benchmark. The multi-objective uncertainty-guided exploration also improves sample efficiency in safety-critical tasks.

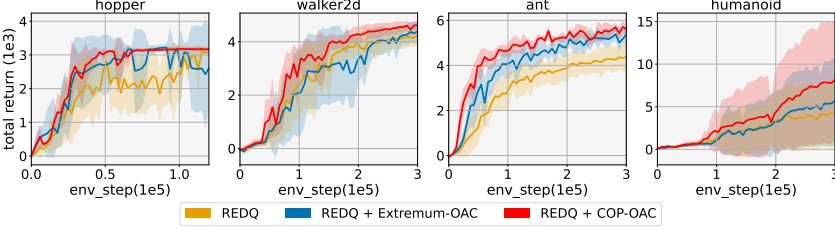

Figure 5: REDQ + COP-OAC vs. REDQ baselines with/without Extremum-OAC enhancement.

## 5.2 VELOCITY-CONSTRAINED MUJOCO (SAFETYVELOCITY-V1)

**Effectiveness in constrained RL**   The results of the SafetyVelocity-v1 benchmark are presented in Figure 6. The numerical results are presented in Appendix E. Compared to on-policy baselines, off-policy methods exhibit significantly higher sample efficiency. Among the off-policy methods, COP-Q shows robust safety performance, maintaining the cost below the threshold throughout the training. SACLag-UCB cannot adhere to the constraints stably. Particularly, even using fewer critics, COP-Q outperforms CAL (UTD=1) in hopper and humanoid, on-par with it in walker2d, and only slightly worse with respect to return in ant. The advantage of COP-Q originates from the prioritized cost learning, which decides the precise adjustment of the Lagrangian multiplier and thus the entire constrained learning process.

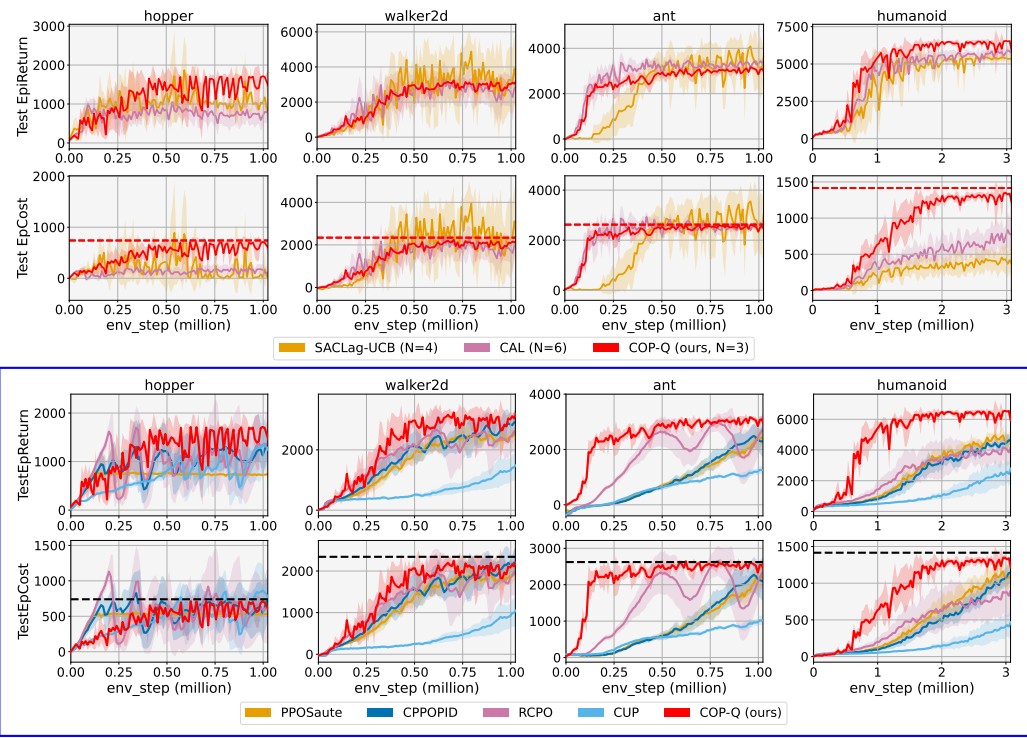

Figure 6: SafetyVelocoty-v1 Benchmark. The top figure compares COP-Q against off-policy baselines (N is the number of critics used). The bottom figure involves on-policy baselines. The horizontal black lines mark the safety threshold.

## 6 CONCLUSIONS

This paper proposed COP-Q, a novel Q-learning method that integrates multi-objective uncertainty into safety-first learning through a Cholesky-based, directed, prioritized structure. COP-Q achieved competitive total returns, rapid safety learning, robust constraint satisfaction, and improved sample efficiency in evaluated MuJoCo and SafetyVelocity-v1 benchmarks. This method introduces minimal computation overhead and is compatible with most Q-learning frameworks, making it promising for many tasks that require learning vector-valued Q-functions.

**Limitations:**   This study has four main limitations. First, the Gaussian assumption on Q-values (expectation of returns), although straightforward, might be oversimplified. Second, epistemic uncertainty (Hüllermeier & Waegeman, 2021) is not disentangled from aleatoric uncertainty, making it difficult to precisely identify truly rare state-action pairs. Third, our experiments did not evaluate the effectiveness of COP-Q in broader settings, such as MORL for approaching the Pareto front. This direction needs more investigation. Fourth, COP-Q cannot compensate for the inherent high estimation biases when facing sparse cost signals, for example, in the SafeNavigation tasks. Extending COP-Q to distributional RL is a plausible future research direction.

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

## A CLIPPING ROTATION MATRIX DETAILS

For a given transformation matrix $\mathbf{C}$ and scalarization vector $\mathbf{u}$, we want to ensure that:

$$\mathbf{u}^\intercal \mathbf{C} \mathbf{R} \mathbf{u} \geq 0. \tag{A.1}$$

If $\mathbf{C}$ is symmetric, the inequality always holds. Otherwise, we first compute $\mathbf{u}^\intercal \mathbf{C} \mathbf{u}$. If positive, simply setting $\mathbf{R} = \mathbf{I}$ (no rotation). If negative, we seek a minimum rotation $\mathbf{R}_m$, such that $\mathbf{u}^\intercal \mathbf{C} \mathbf{R} \mathbf{u} = 0$. This $\mathbf{R}_m$ serves as the clipping operation as illustrated in the right plot in Figure 2. A computationally efficient way to compute $\mathbf{R}_m$ is to use the *Rodrigues formula* (Dai, 2015).

We note:

$$\mathbf{p} = \mathbf{C}^T \mathbf{u}, \quad a = \mathbf{u}^\intercal \mathbf{p}, \quad \rho = ||\mathbf{p}|| > 0, \tag{A.2}$$

also,

$$\mathbf{v} = \frac{\mathbf{p} - a\mathbf{u}}{\sqrt{\rho^2 - a^2}}, \quad \mathbf{K} = \mathbf{v}\mathbf{u}^\intercal - \mathbf{u}\mathbf{v}^\intercal. \tag{A.3}$$

Then the clipping rotation matrix is:

$$\mathbf{R}_m = \mathbf{I} - \frac{a}{\rho}\mathbf{K} + (1 - \sqrt{1 - \frac{a^2}{\rho^2}})\mathbf{K}^2 \tag{A.4}$$

It is easy to confirm that this is an isometry.

In practice, the condition $\mathbf{u}^\intercal \mathbf{C} \mathbf{u} < 0$ is rarely triggered because this means that the estimates of different objectives are strongly negatively correlated with very different scales of marginal variance. Geometrically, the confidence region becomes a thin ellipsoid with a large condition number. Nevertheless, adding this clipping operation theoretically guarantees that the total objective is never compromised, which is important for mitigating overestimation. Also, it extends the scalar clipping operator to a multi-objective form. We summarize and compare single-objective and multi-objective confidence bounds in Table.A.1.

Table A.1: Comparison between single-objective and multi-objective confidence bounds

|  | Single-objective |  | Multi-objective (proposed) |
|---|---|---|---|
| Form | $\mu \pm \sqrt{\chi_1^2(p)} \times \sqrt{\sigma^2} \times 1$ | $\rightarrow$ | $\boldsymbol{\mu_Q} \pm \sqrt{\chi_N^2(p)} \times (\mathbf{C}\mathbf{R})\mathbf{u}$ |
| Scale | $\chi_1^2(p)$, two end points |  | $\chi_N^2(p)$, an ellipsoid |
| Uncertainty | $\sigma$, square root of variance | $\rightarrow$ | $\mathbf{C}$, factorization of covariance matrix |
| Scalarization | 1, trivial coefficient |  | $\mathbf{u}$, a direction vector |
| clipping | 1, trivial coefficient |  | $\mathbf{R}$, a rotation |
| Meaning | quantification + clip | $\rightarrow$ | quantification + projection + rotation |

## B COMPARISON OF PROJECTION METHODS

Table B.1: Properties of different projections ($\boldsymbol{\Lambda}$: diagonal eigenvalue matrix $\mathbf{v}$: eigenvectors)

| Method | form of $\mathbf{C}$ | form of $\mathbf{R}$ | Correlations | Priorities |
|---|---|---|---|---|
| Cholesky | $\mathbf{L}$ | equation A.4 | ✓ | ✓ |
| Eigen-decomposition | $\mathbf{v}^\intercal \boldsymbol{\Lambda}^{1/2} \mathbf{v}$ | $\mathbf{I}$ | ✓ | ✗ |
| Extremum | $\boldsymbol{\Sigma}/\sqrt{\mathbf{u}^\intercal \boldsymbol{\Sigma} \mathbf{u}}$ | $\mathbf{I}$ | ✓ | ✗ |
| Diagonal | $\boldsymbol{\Lambda}^{1/2}/\|\mathbf{u}\|_2$ | $\mathbf{I}$ | ✗ | ✗ |

Table.B.1 further compares the properties of other projection methods that might be useful for other types of tasks. Except for Cholesky factorization, other methods have symmetric transformation matrices; thus, no rotation is needed.

These projected lower bound points are marked and compared in Figure B.1. The opposite directions are upper bounds. No matter what the covariance matrix is, the Cholesky projection is always on the conservative side of safety, if safety is ranked first, while other methods may shift to the optimistic side of safety, depending on the orientation of the ellipsoid. Note that, except for Cholesky factorization, the transformation matrices $\mathbf{C}$ in other projections are all symmetric, which means they do not have priorities across objectives.

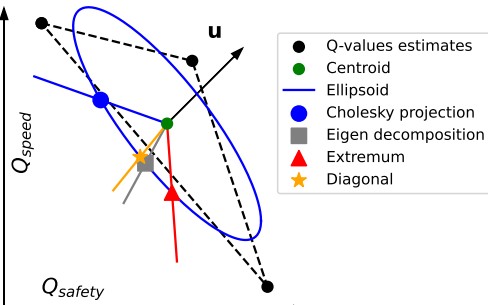

Figure B.1: Lower confidence bounds using different projection methods.

## C  BRAX-MUJOCO ENVIRONMENT

**Standard MuJoCo with multiple objectives**  The 5 locomotion tasks include 1 single-legged robot (hopper), 2 bipedal robots (walker2d and humanoid), and 2 quadrupedal robots (ant and halfcheetah). Their configurations can be found on the website of Gymnasium documentation (https://gymnasium.farama.org/environments/mujoco/).

For hopper, walker2d, humanoid, and ant, there are 3 rewards:

- **Safety:** If the robot's state is in the safe range, e.g., not falling, get a constant reward +5 (for humanoid) or +1 (for hopper, walker2d, and ant) every time step; otherwise, get 0, and the episode terminates immediately.

- **Velocity:** The velocity of the robot along the given positive x-axis direction. This reward can be positive or negative (opposite direction).

- **Control cost:** A non-positive reward that penalizes excessive or inefficient use of control inputs. The action of each actuator is $u_i \in [-1, 1]$, and the control cost is:

$$\text{control cost} = -\sum_i u_i^2. \tag{C.1}$$

For halfcheetah, there is no safety reward, and the episode only ends when reaching maximum simulation steps (1000 for all tasks). To focus on the exploration strategy of the policy itself, the reset noise of initial states is set to 0.005 for all tasks, which is a small perturbation. Brax-MuJoCo provides different simulation backends. We use the "mjx" backend for humanoid and the default "generalized" backend for other robots.

**SafetyVelocity-v1**  We use similar settings as in Safety-Gymnasiums (Ji et al., 2023), but implement them in Brax for consistency and training speed concerns. Here we select 4 robots, hopper, walker2d, ant, and humanoid, because they share the same reward structure:

$$r_t = w_h \times r_{\text{health}} + w_v \times r_{\text{velocity}} - w_c \times r_{\text{ctrl}}, \tag{C.2}$$

in which $w_h$, $w_v$ and $w_c$ are positive scalarization weights. Following the formulation in the CVPO paper (Liu et al., 2022) and CAL (Wu et al., 2024), the cost limit is set on the episodic velocity returns. Used scalarization weights for the reward and the cost thresholds are listed in Table C.1.

For every simulation step, the agent receives the reward $r_t$ and a cost $c_t$, their Q-functions are denoted as $Q_r^\pi(s, a)$ and $Q_c^\pi(s, a)$, respectively. If the cost limit on $Q_c^\pi(s, a)$ is noted as $d$ (which can

Table C.1: Weight coefficients and velocity threshold for SafetyVelocity-v1

| **ROBOT** | $(w_h, w_v, w_c)$ | Episode velocity return threshold |
|---|---|---|
| hopper | (1, 1, 0.001) | 740.2 |
| walker2d | (1, 1, 0.001) | 2341.5 |
| ant | (1, 1, 0.5) | 2622.2 |
| humanoid | (5, 1.25, 0.1) | 1411.9 |

be estimated from the episode cost threshold using the formula in CVPO paper (Liu et al., 2022), approximately 1/10 of the episode cost threshold when using 0.99 as discount factors), constrained RL considers a constrained optimization problem:

$$\max_{\pi} \mathbb{E}_{s \sim \rho_\pi, a \sim \pi(\cdot|s)}[Q_r^\pi(s, a)], \quad \text{s.t.} \quad \mathbb{E}_{s \sim \rho_\pi, a \sim \pi(\cdot|s)}[Q_c^\pi(s, a)] \leq d, \tag{C.3}$$

where $\rho_\pi$ is the state density function of the policy $\pi$. The primal-dual approach converts it into the following dual form:

$$\arg \min_{\lambda > 0} \mathbb{E}_{s \sim \rho_\pi, a \sim \pi(\cdot|s)}[Q_r^\pi(s, a) - \lambda(Q_c^\pi(s, a) - d)], \tag{C.4}$$

$$\arg \min_{\lambda > 0} \lambda \times (d - \mathbb{E}_{s \sim \rho_\pi, a \sim \pi(\cdot|s)}[Q_c^\pi(s, a)]), \tag{C.5}$$

In learning, we update the policy $\pi$ and Lagrangian multiplier $\lambda$ iteratively. So, the un-normalized scalarization weights $(1, -\lambda)$ keep changing during training.

# D   ALGORITHMIC AND IMPLEMENTATION DETAILS

This appendix explains the details and hyperparameters of the proposed COP-Q, and also the implementation of baselines.

**Cholesky factorization computation**   For Jax and PyTorch users, we do *not* recommend using the default Cholesky factorization functions (`jnp.linalg.cholesky`, `jax.scipy.linalg.cholesky`, `torch.linalg.cholesky`, or the related `chol_factor` + `chol_solve`). These functions cannot deal well with the degenerated cases with large conditional numbers (almost co-linear or co-planar). And it is important to note that adding diagonal jitters to the covariance matrix cannot solve this problem stably, especially using the default float32 precision in Jax. Training sometimes gives NaN results.

Two solutions are tested in our experiments. If the number of objectives $N$ is not very large, writing the analytical form of Cholesky decomposition and fixing the jitter inside the function manually works well. For example, add 1e-7 for float32 precision when applying the square root. If $N$ is large, like 10-50 in multi-task reinforcement learning, a more complex version called *pivoted Cholesky decomposition* works stably for a nearly low-rank, ill-conditioned covariance matrix. TensorFlow provides such functionality. For readers using other frameworks, refer to the implementations in Harbrecht et al. (Harbrecht et al., 2012).

**FAMO:**   The critic update uses the Fast Adaptive Multi-task Optimization (FAMO) (Liu et al., 2023) to balance the learning of all Q-functions, avoiding that one objective dominates the gradient and other objectives are ignored. We choose FAMO because it has $\mathcal{O}(1)$ complexity, and is easy to implement. The idea is to adaptively adjust the weights of the mean-squared error of different objectives in the critic update. We refer the readers to the original paper for more details. The only hyperparameter of FAMO is the learning rate and the weight decay of the AdamW weight optimizer (Loshchilov & Hutter, 2017). Here we choose the recommended values.

**OAC:**   Optimistic actor-critic (Ciosek et al., 2019) is UCB-like exploration strategy for continuous control tasks. Besides the target policy $\pi_t(s) = \mathcal{N}(a|\mu_t, \Sigma_t)$, OAC induces an additional exploration policy $\pi_e(s) = \mathcal{N}(a|\mu_e, \Sigma_e)$. OAC trains the target policy in offline updates, while using the

Table D.1: FAMO hyperparameters of AdamW adaptive weight optimizer

| Parameters | Value |
|---|---|
| learning rate | 0.025 |
| weight decay | 0.001 |

exploration policy to interact with the environment to collect data. The exploration policy has the following action distribution:

$$
\begin{cases}
\mu_e & = \mu_t + \sqrt{2\delta} \cdot \dfrac{\Sigma_t \nabla_a \hat{Q}_{\mathrm{UB}}(s,a)\big|_{a=\mu_t}}{\left\| \nabla_a \hat{Q}_{\mathrm{UB}}(s,a)\big|_{a=\mu_t} \right\|_{\Sigma_t}}, \\
\Sigma_e & = \Sigma_t
\end{cases}
\tag{D.1}
$$

where $|| \cdot ||_{\Sigma_t}$ is the Mahalanobis norm. For multi-objective Q-learning, the upper bound of Q-values is given by equation 15. So the action sampling is biased toward the direction with a higher optimistic upper bound. Therefore, OAC has two hyperparameters, $\sqrt{2\delta}$, which is the KL divergence constraint between the target policy and the exploration policy, and the exploration step length $d_m$ (Mahalanobis distance). In the original single-objective OAC, the exploration bonus is the standard variance. In this study, we use the recommended values in the original paper.

Table D.2: OAC hyperparameters

| Parameters | Value |
|---|---|
| shift multiplier $\sqrt{2\delta}$ | 6.86 |
| exploration step length $\beta$ | 4.66 |

**COP-Q Algorithm framework for standard MuJoCo benchmark:** The proposed COP-Q is implemented in TD3 for the standard MuJoCo benchmark, with an optional OAC exploration module. The framework is summarized in Alg.1. The used hyperparameters of TD3 (for all non-REDQ methods) are listed in Table.D.3. Most hyperparameters are the same as the Optimistic Actor-Critic paper (Ciosek et al., 2019). Only the minimum buffer length (collecting how many samples before starting the network update) and batch size are changed. We use a larger initial buffer size for stabilizing multi-objective Q-values learning, and also a larger batch size of 256 for the same reason. Using a smaller minimum buffer size, such as 10,000 in OpenAI Spinning up, may slow down the convergence in our trials.

For REDQ-based methods, most hyperparameters are the same. Only the update-to-date (UTD) ratio is increased to 20, and the actor delay also increases to 20. That means after collecting 1-step transition, we apply randomized critic updates 20 times, and then apply 1 actor update, and then get the next-step transition from the simulation.

**COP-Q Algorithm for SafetyVelocity-v1:** COP-Q is implemented based on CAL (Wu et al., 2024), which is a Soft Actor-Critic primal-dual constrained RL framework. The shared hyperparameters remain the same as the original CAL paper, and the COQ-Q specific hyperparameters are listed in Table D.3, the SAC part. The only difference is that we put the cost objective in the first place and use 3 critics. Each critic estimates both reward and cost objectives. We refer the readers to Wu et al. (2024) and its open-source code for more details. Note that the policy update frequency and target update frequency are decreased to 2 for stability (also applied to all off-policy baselines). Also, it is useful to note that the entropy term is only added to the reward objective, not to the cost objective. This is a standard setting in most constrained RL methods (Ji et al., 2024).

**Implementation of baselines in SafetyVelocity-v1** CAL (Wu et al., 2024) is implemented based on its official code, only dropping the randomized ensemble part and set UTD=1 for fairness. SACLag-UCB is also described in CAL. We use their settings and implement them based on the SACLag algorithm provided by OmniSafe (Ji et al., 2024), a constrained (safe) RL benchmark platform.

---

**Algorithm 1** COP-Q based on TD3, with optional COP-OAC module

---

**Input and initialization:** policy network $\pi_\theta(s)$, $N+1$ critic networks $\{q_{\psi_i}\}_{i=1}^{N+1}$, replay buffer $\mathcal{D}$, normalized scalarization vector $\mathbf{u}$, Mahalanobis distance for exploration $\chi_N^2(p_e)$, rank the objectives in order based on priorities.
**repeat**
    Observe State $s_t$,
    **if** Use OAC **then**
        Compute $\hat{Q}_{UB}(s,a)$ from critics using equation 15
        Replace $\mu_t$ in Eq,equation D.1 by $\mu_\theta(s_t)$, compute $\mu_e$
        select action $a_t = \text{clip}(\mu_e + \epsilon, a_{\text{lower}}, a_{\text{upper}})$, where $\epsilon \sim \mathcal{N}(0, \sigma_t^2 \mathbf{I})$.
    **else**
        select action $a_t = \text{clip}(\mu_\theta(s_t) + \epsilon, a_{\text{lower}}, a_{\text{upper}})$, where $\epsilon \sim \mathcal{N}(0, \sigma_t^2 \mathbf{I})$.
    **end if**
    Execute $a_t$, observe next state $s_{t+1}$ and reward vector $\mathbf{r}_t$
    Store the transition $(s_t, a_t, \mathbf{r}_t, s_{t+1})$ in $\mathcal{D}$
    **if** critic update **then**
        Sample a random batch of transitions from $\mathcal{D}$
        Compute TD target $\mathbf{y_t}$ using equation 13
        Update critics by minimizing weighted TD errors through FAMO
    **end if**
    **if** actor update **then**
        Sample a random batch of transitions from $\mathcal{D}$
        Compute actor's objective to maximize using equation 14.
        Update the target policy network.
    **end if**
**until** Convergence

---

SACLag-UCB uses 2 reward critics and 2 cost critics to estimate both objectives independently and conservatively. It is especially useful to note that, in constrained RL, due to the negative coefficient of the cost objective, a higher bound of cost is conservative. For COP-Q, the projection has already taken the scalarization weights $(1, -\lambda)$ into consideration. So, no additional changes are needed. For on-policy baselines, all implementations use the same 1M steps hyperparameters recommended by OmniSafe (Ji et al., 2024). We refer the readers to their open-source code for details.

**Compute resources** Brax is fully jaxed, so all experiments are conducted on the internal GPU clusters. For each training, 1 or 2 Nvidia V100 with 32GB of memory are used.

# E ADDITIONAL RESULTS AND ANALYSIS

**Quantitative results of the main benchmark of standard MuJoCo** Table.E.1 lists the average of the best recorded total return in each training for the main benchmark. Note that this may seem intuitively different from Figure 3 because oscillations can significantly influence the results. For example, on walker2d, extremum projection seems better than scalarization double-Q. However, in this Table, scalarization double-Q has a higher score because the training curves have some unstable high spikes. The best recorded total return in each training is sensitive to these spikes. Table.E.2, in contrast, lists the maximum of the averaged training curves. Nevertheless, both tables are less informative than the Training curves.

**Quantitative results of the SafetyVelocity-v1 benchmark** Table E.3 shows the results of the SafetyVelocity-v1 benchmark, including off-policy baselines and on-policy baselines. The bold numbers are marked based on the best return performance while adhering to the cost constraints. For example, in ant, although SAClag-UCB achieves the highest return, the cost exceeds the threshold. So, the highest return does not count.

Table D.3: COP-Q hyperparameters

| Parameters | Value |
|---|---|
| Shared parameters | |
| optimizer | Adam |
| critic learning rate | 0.0003 |
| actor learning rate | 0.0003 |
| target smoothing coefficient | 0.005 |
| actor delay | 2 |
| batch size | 256 |
| discount | 0.99 |
| replay buffer size | 1,000,000 |
| critic network size | (256, 256) |
| actor network size | (256, 256) |
| nonlinearity | ReLU |
| TD3 for standard MuJoCo | |
| target update frequency | 1 |
| gradient-to-data ratio | 1 |
| action noise | 0.1 |
| policy noise | 0.2 |
| noise clipping range | 0.5 |
| FAMO | True |
| Primal-dual SAC for SafeVelocity-v1 | |
| target update frequency | 2 |
| gradient-to-data ratio | 2 |
| entropy coefficient auto-tuning | True |
| entropy coefficient learning rate | 5e-4 |
| Lagrangian initial value | 1 |
| Lagrangian learning rate | 5e-4 |
| FAMO | False |

Table E.1: Average of the best total return in each training over 5 runs

| Methods | hopper | walker2D | humanoid | ant | halfcheetah |
|---|---|---|---|---|---|
| Scalarization | 3164 | 4686 | 13120 | 5538 | 16650 |
| Independent | 2604 | 428 | 15819 | 39 | 17339 |
| Extremum | 3048 | 4405 | 13252 | 5554 | 16064 |
| COP-Q (ours) | **3171** | **5254** | **18365** | **5825** | **18463** |

Table E.2: Maximum of the total return in training averaged over 5 runs

| Methods | hopper | walker2D | humanoid | ant | halfcheetah |
|---|---|---|---|---|---|
| Scalarization | **3136** | 4347 | 12867 | 5468 | 16579 |
| Independent | 2549 | 411 | 15520 | 34 | 17158 |
| Extremum | 3034 | 4378 | 13186 | 5483 | 15822 |
| COP-Q (ours) | 3092 | **5207** | **18210** | **5810** | **18372** |

## F THE USE OF LARGE LANGUAGE MODELS (LLMS)

LLMs are used for polishing writing only, such as selecting the proper words or correcting grammar mistakes.

Table E.3: Benchmark of SafetyVelocity-v1 (* means constraint violation)

| Env(steps) | Hopper(1M) | | Walker2d(1M) | | Ant(1M) | | Humanoid(3M) | |
|---|---|---|---|---|---|---|---|---|
| Metric (constraint) | Reward | Cost (740) | Reward | Cost (2342) | Reward | Cost (2622) | Reward | Cost (1415) |
| **COP-Q** | **1452** ±471 | 592 ±194 | **3198** ±114 | 2214 ±90 | 2924 ±474 | 2415 ±350 | **5926** ±758 | 1216 ±149 |
| CAL | 910 ±261 | 83 ±85 | 2888 ±1002 | 2045 ±692 | **3363** ±126 | 2594 ±116 | 5742 ±440 | 773 ±236 |
| SACLag-UCB | 846 ±372 | 32 ±12 | 3000 ±1601 | 2291 ±1262 | 3840* ±252 | 3256* ±282 | 5371 ±226 | 372 ±182 |
| CPPOPID | 1390* ±306 | 747* ±311 | 2917 ±427 | 2200 ±369 | 2292 ±584 | 2097 ±547 | 4629 ±180 | 1142 ±109 |
| RCPO | 1362 ±696 | 646 ±519 | 2563 ±712 | 1942 ±572 | 2747 ±465 | 2379 ±377 | 4163 ±894 | 901 ±321 |
| PPOSaute | 740 ±10 | 536 ±3 | 2508 ±214 | 1940 ±176 | 2378 ±361 | 2173 ±336 | 4678 ±661 | 1102 ±162 |
| CUP | 1227 ±302 | 739 ±172 | 1497 ±516 | 1054 ±383 | 1252 ±140 | 1019 ±131 | 2783 ±883 | 470 ±175 |

