# OpenReview forum: "COP-Q: Safety-First Reinforcement Learning with Cholesky Ordered Projection"
_ICLR.cc/2026/Conference — Submitted to ICLR 2026_

### Official Review · Reviewer_xEpw · 2025-10-21

**Soundness:** 2
**Presentation:** 3
**Contribution:** 3
**Rating:** 4
**Confidence:** 3

**Summary:**

In multi-objective RL, uncertainties of Q functions for different objectives may become intertwined, introducing extra challenges. Existing methods treat uncertainty in each objective independently or collapse them into a scalarized dimension. In this work, the authors introduce Cholesky Ordered Projection Q-learning (COP-Q), using full multi-objective uncertainty by covariance matrix factorization. Extensive experiments on standard MuJoCo and velocity-constrained SafetyVelocity-v1 benchmarks, demonstrating the effectiveness of COP-Q.

**Strengths:**

- Figures 1 and 2 clearly show the insights and contributions of this work.

- It is interesting and novel to introduce Cholesky Ordered Projection into RL.

- Extensive experiments in both standard settings and safe settings show that COP-Q can handle different objectives in the same time.

**Weaknesses:**

- I'm curious that if the Scalarization weight is fixed, what is the major difference between multi-objective RL and single-objective RL (R = u^T r)?

- As mentioned in lines 146-149, the Scalarization weight u might be fixed or changed. However, I find that the main algorithm seems designed for fixed u, what about handling the changing u (if I have any misunderstanding, please point it out)?

- In Fig. 1, the authors mention that the uncertainty of Q total may be low, but the uncertainty of each Q value may be high. Are there any theoretical or experimental observations supporting this insight?

Overall, I think the idea of this work is novel, but there are still some concerns. I'd like to adjust my score if the authors can address my concerns.

**Questions:**

See weaknesses above.

---

> ### Author Response · Authors · 2025-11-22
> **Official responses of Submission24260 to Reviewer xEpw - part 1**
>
> We sincerely appreciate your careful review of our manuscript and your relevant, constructive comments. Our responses to each raised concern are given below.
>
> &nbsp;
>
> **Q1. The major difference between multi-objective RL and single-objective RL when the scalarization weight is fixed**
>
> In short, the difference is **whether learning or monitoring the Q-function of each objective in training and deployment.**
>
> The "multi-objective" and "single-objective" RL that the reviewer mentioned here actually correspond to the **Scalarization of Expected Return (SER)** and **Expectation of Scalarized Return (ESR)** approaches in MORL, as explained in Section 2, lines 104-109:
>
> - **SER**: Learning the Q-function of each objective explicitly. Critics give a vector of Q-values, then the scalarization weight converts the Q-vector into the total return.
>
> - **ESR**: Learning the total Q-function directly. Critics give the total Q-value conditioned on the scalarization weight (for example, [1]).
>
> Although SER and ESR indeed give *quantitatively identical* final objective for the actor under linear scalarization, they radically differ in critic updates and deployment:
>
> **First, ESR does not allow monitoring of the safety-critical objective.** Therefore, in many real-world applications, such as robot manipulation and autonomous driving, the critics only evaluate the total return and its uncertainty, thus being blind to potential risk.
>
> **Second, SER induces complex gradient conflicts and correlated uncertainty.** Learning multiple objectives explicitly makes SER not as simple and neat as ESR--it induces additional complexity in critic updates as a cost.
>
> - **Gradient conflict:** As now each critic has multiple outputs and multiple loss functions, optimizing the summed loss may harm the accuracy of some objectives. In this paper, as explained in Appendix D, we use the existing FAMO method [2] to ensure balanced learning of multiple Q-functions.
>
> - **Correlated uncertainty:** As explained in Fig.1, negatively correlated objectives induce conflicting estimation biases. When the safety-critical objective is overestimated, the agent will act unsafe. Therefore, the core of this study is introducing multi-objective confidence bounds and employing Cholesky decomposition to ensure the priority of safety while avoiding being over-conservative about other objectives.
>
> In summary, as explained, **this study is in the category of SER, and it aims to overcome the challenge of correlated uncertainty.** Using single-objective RL cannot make the agent be safety-aware in deployment.
>
> To make this point explicit, we modify the relevant sentences in the related work section:
>
> - line 107-107: *"ESR is not relevant to this study, because it converts the problem into a single-objective task that is not aware of the risk."*
>
> [1] *Diederik M Roijers, Denis Steckelmacher, and Ann Nowe. Multi-objective reinforcement learning for the expected utility of the return. In Proceedings of the Adaptive and Learning Agents workshop at FAIM, volume 2018, 2018.*
>
> [2] *Bo Liu, Yihao Feng, Peter Stone, and Qiang Liu. Famo: Fast adaptive multitask optimization. Advances in Neural Information Processing Systems, 36:57226–57243, 2023.*
>
> &nbsp;
>
> **Q2. About the changing scalarization weight**
>
> When $u$ is changing during training, we directly **use the latest updated value of $u$**. Nothing else changes for our approach.
>
> In fact, the constrained RL task in Section 5.2 (SafetyVelocity) is such an example. The problem formulation is given in Appendix C (Eq C.3-C.5). In the used primal-dual approach, the Lagrangian multiplier $\lambda$ is updated during training. The weight coefficients for reward and cost are 1 and $-\lambda$, respectively. In learning, we update the policy and $\lambda$ iteratively based on whether the cost violates the constraint. So, the un-normalized scalarization weight $(-\lambda, 1)$ keeps changing during training. The only thing that we need to do is put the cost priority ahead of reward (for safety) and normalize the length of $(-\lambda, 1)$ to 1.
>
> To clarify this point, we add the following explanations in the introduction part of the experiment section:
>
> - line 292-294: *"The changing Lagrangian multiplier thus composes a changing scalarization weight $u$. We therefore use the latest in-place scalarization weight in each update iteration during training."*

---

> ### Author Response · Authors · 2025-11-22
> **Official responses of Submission24260 to Reviewer xEpw - part 2**
>
> **Q3. In Fig. 1, the authors mention that the uncertainty of Q total may be low, but the uncertainty of each Q value may be high. Are there any theoretical or experimental observations supporting this insight?**
>
> We clarify that the "uncertainty" in our discussion refers to the covariance matrix of the Q-values estimated by an ensemble of critics. We hypothesize that when a single critic network predicts both Q-x and Q-y simultaneously, gradient conflicts and challenging optimization dynamics can cause the uncertainties of the individual Q-values to become **negatively correlated**. This negative correlation, in turn, causes the uncertainty in the total Q-value to be smaller than the uncertainty in each objective.
>
> Since the covariance matrix was not explicitly stored during training, providing direct evidence requires re-running some of the experiments. We are currently doing so and will report the results in a follow-up comment in this thread during the rebuttal period. We are sharing this preliminary explanation now to facilitate a timely and productive discussion.
>
> &nbsp;
>
> **Overall, I think the idea of this work is novel, but there are still some concerns. I'd like to adjust my score if the authors can address my concerns.**
>
> Thank you for your review. We hope that our responses address your concerns. If you have any more questions, we would be happy to discuss them.

---

> > ### Author Response · Authors · 2025-11-24
> > **Additional results for our response to Q3.**
> >
> > We have conducted additional experiments to directly demonstrate that the uncertainty in the total Q-value can be lower than the uncertainty of each objective.
> >
> > We re-ran training for *hopper*, *walker2d*, and *ant* using the same 5 random seeds, and explicitly recorded the covariance matrix of the Q-values during the first 0.5M steps for *hopper* and 1M steps for *walker2d* and *ant*. Through the training, we computed the ratio (%) of:
> >
> > - (i) cases where the standard deviation of $Q_{\text{total}}$ is smaller than at least one objective’s standard deviation;
> >
> > - (ii) cases where it is smaller than all the objectives’ standard deviations.
> >
> > We also consider two ways to define $Q_{\text{total}}$:
> >
> > &nbsp;
> >
> > **(1) the uncertainty in the directly summed total objective using unnormalized scalarization weights**
> >
> > In this way, $Q_{\text{total}}$ is formed using the raw scalarization weights $u$ (not normalized to unit length). The results are given below:
> >
> > | Ratio (%, mean ± std)            | Hopper | Walker2d | Ant |
> > |----------------------|---------------------|------------------------|------------------|
> > | $\sigma_{\text{total}}$ < std of at least one objective     | 37.2 ± 5.1          | 39.7 ± 2.0             | 80.4 ± 2.4       |
> > | $\sigma_{\text{total}}$ < stds of all objective      | 4.8 ± 0.8          | 5.7 ± 0.1             | 7.0 ± 0.8       |
> >
> > &nbsp;
> >
> > **(2) the uncertainty in the total objective using normalized scalarization weights**
> >
> > Here, we first normalize the scalarization vector $u$ to unit length before forming $Q_{\text{total}}$. The results are given below:
> >
> > | Ratio (%, mean ± std)            | Hopper | Walker2d | Ant |
> > |----------------------|---------------------|------------------------|------------------|
> > | $\sigma_{\text{total}}$ < std of at least one objective     | 92.3 ± 1.8         | 93.8  ± 1.8         | 99.9 ± 0.1       |
> > | $\sigma_{\text{total}}$ < stds of all objective      | 21.9 ± 3.8          | 21.7 ± 0.6             | 27.0 ± 2.3      |
> >
> > &nbsp;
> >
> > These results directly confirm that:
> >
> > - Negative correlations between objectives are common during MuJoCo training,
> >
> > - The uncertainty (standard deviation) of $Q_{\text{total}}$ can frequently be lower than that of individual objectives, often lower than all of them, especially when using normalized scalarization weights.
> >
> > We believe these additional experiments address your concern in Q3

---

> > > ### Comment · Reviewer_xEpw · 2025-11-25
> > >
> > > Thanks for your response. Most of my concerns are addressed, but I still have some questions:
> > >
> > > - I understand that ESR is one kind of method for handling MORL and is similar to single-objective RL. Also, I understand that ESR can not handle safety RL, which needs to keep the safety constraint controlled. My question is, if the u in your Eq. 2 is fixed, is ESR optimal for handling MORL? In other words, in these cases (u is fixed), what are the advantages of SER and COP-Q?
> > >
> > > - By the way, in constrained RL, like safe RL [1-3], if we use the Lagrange multiplier method to transform it into an unconstrained optimization problem, then the coefficient lambda needs to be optimized, which means that u can be changed. I am curious whether, in this case, ESR is optimal?
> > >
> > > I'd like to follow the discussion and raise my scores if my concerns are addressed.
> > >
> > > Reference:
> > >
> > > [1] Safedreamer: Safe reinforcement learning with world models
> > >
> > > [2] Towards Safe Reinforcement Learning via Constraining Conditional Value at Risk
> > >
> > > [3] CVaR-constrained policy optimization for safe reinforcement learning

---

> > > > ### Author Response · Authors · 2025-11-25
> > > >
> > > > Thank you for your follow-up questions and clarifications. Our responses are as follows.
> > > >
> > > > **(1) Advantages of SER / COP-Q when $u$ is fixed**
> > > >
> > > > If “optimality” refers to the *total return* for a fixed $u$, then ESR, SER, and COP-Q are theoretically equivalent after convergence: they all optimize the same scalarized objective. In practice, however, **ESR is usually more sample-efficient than SER/COP-Q** for optimizing total return, because it solves a single-objective problem with a neat optimization landscape (no gradient conflict, single scalar label).
> > > >
> > > > The picture changes once safety is important. COP-Q uses Cholesky-based prioritization to always place safety first in learning. This design leads to faster and more stable convergence of the safety objective than ESR, where the safety learning is implicitly mixed into a single scalar.
> > > >
> > > > In short, if we only care about total return, ESR is effectively “optimal” and often more sample-efficient. In contrast, COP-Q is designed to optimize safety first, even when  $u$ is fixed.
> > > >
> > > > **(2) ESR in primal-dual constrained RL**
> > > >
> > > > To be frank, we are not sure how to apply ESR in primal-dual constrained RL. The dual update of Lagrangian multiplier $\lambda$ depends explicitly on the cost:
> > > >
> > > > $\text{arg min}\ \lambda\times(d - Q_{\text{cost}}),$
> > > >
> > > > in which $d$ is the threshold. In on-policy methods, one can use the observed cost in data collection directly to replace $Q_{\text{cost}}$ because the data-collection policy and the optimized policy are identical. In off-policy safe RL, however, the data-collection policy differs from the target policy, so we need to use the critics to estimate the cost under the current policy and adjust $\lambda$ correctly.
> > > >
> > > > For this reason, it is not clear to us how ESR would fit into the primal–dual framework. To the best of our knowledge, we are not aware of any off-policy constrained RL works that use ESR. Most of them use separate critics for reward and cost.
> > > >
> > > > &nbsp;
> > > >
> > > > We hope these clarifications address your follow-up questions. If any part remains unclear, we would be happy to discuss further.

---

> > > > > ### Comment · Reviewer_xEpw · 2025-11-25
> > > > >
> > > > > Thanks for your response, which has addressed all my concerns. I have raised my score to 6 and think the idea of introducing Cholesky Ordered Projection into MORL is interesting and novel. However, I hope the authors to better clarify the relationship between ESR and SER (under fixed u or changing u), and safety RL and safety constrained RL (in the view of MORL), in the revised version, which can better position this work and help readers to understand the contribution of this work.

---

### Official Review · Reviewer_cZ7N · 2025-10-26

**Soundness:** 2
**Presentation:** 1
**Contribution:** 2
**Rating:** 2
**Confidence:** 2

**Summary:**

The paper aims to quantify uncertainty in multi-objective reinforcement learning.
This is realized using Cholesky factorization in the multi-objective Q-space.
By having a richer representation of the uncertainty, the overall performance is slightly improved.

**Strengths:**

- The paper explicitly considers the uncertainty in the multi-objective Q-space.
- The approach follows a hierarchical schema, where certain objectives (e.g., safety) are prioritized over others.
- Experiments indicate a slight improvement over compared methods.

**Weaknesses:**

- The paper is extremely difficult to follow, in particular, Sec. 4.1:
The applied steps (Eq. 6-8) require more explanations, clearly motivating what the goal of the subsequent transformations is, and properly explaining all variables. e.g., $R$. Similar holds for Eq. (10), where $C_{clip} = CR$ is introduced to drop $R$, but never used again. One can assume that $C_{clip}$ became $L_{clip}$ through Sec. 4.2, but this connection can be made clearer, or avoid dropping R altogether.
- The paper also does not state clearly its assumptions / properly define the variables. For example, the paper says that (A.1) always holds if $C$ is symmetric. However, for $C=-I$, this clearly does not hold for any $u$, indicating that there are certain restrictions on which values $C$ can take.
- It is unclear what is meant by the priorization of the objectives in Sec. 4.2. Are these given by the scalarization vector $u$?
- The term "uncertainty" is not well defined.

**Questions:**

- Can you explain the term "the level of optimism when facing uncertainty" more intuitively than simply stating (13)?
- Does it matter how $u$ is set to determine the priorization of the objectives and vice-versa?
- Can you quantify the "minimal computational overhead" of your approach?

---

> ### Author Response · Authors · 2025-11-22
> **Official responses of Submission24260 to Reviewer cZ7N - part 1**
>
> We thank the reviewer for carefully evaluating our submission. The raised questions mainly concern clarifications of the details of our proposed method. Our point-by-point responses are provided below.
>
> &nbsp;
>
> **Q1.1 The paper is extremely difficult to follow, in particular, Sec. 4.1: The applied steps (Eq. 6-8) require more explanations, clearly motivating what the goal of the subsequent transformations is, and properly explaining all variables. e.g., R.**
>
> We appreciate the reviewer’s comment and agree that Sec. 4.1 can be further clarified. The overall goal is stated by the first sentence:
>
> - "*generalize the confidence bounds of the Q-value from a scalar (for single-objective RL) to a vector (for multi-objective RL)*".
>
> In the multi-objective case, a confidence interval is no longer a point or a line segment but an ellipsoid in the multi-dimensional Q-space, and we also have a scalarization vector $u$. **Intuitively, we want to construct a point on this ellipsoid considering the direction of $u$**, which serves as a multi-objective lower confidence bound. Therefore, Eq. (6) introduces a linear transformation $A$ applied to $u$. To ensure that the projected point lies on the ellipsoidal confidence boundary, we need to seek the general form (solution) of $A$.
>
> Eqs. (7)–(8) then solve for $A$ in terms of the given covariance matrix. This is where the factorization of the covariance matrix comes in. As stated:
>
> - *"This family of solutions is informative because $C^{-1}$ “whitens” the Gaussian to $N (\mu_Q, \chi^2N(p)I)$ (diagonal). Conversely, $C$ re-injects full inter-objective covariances into confidence bounds."*
>
> Geometrically, $C^{-1}$ transforms the ellipsoid into a sphere ("whitening"), and $C$ maps it back, ensuring that the final bound correctly reflects the full covariance structure. This process leads to Eq. (9), which gives the general solution for $A$, and thus the multi-objective lower confidence bound.
>
> The matrix $R$ appears when solving the right-hand side of Eq. (7). As explained, the solution for $A$ is not unique; any isometry corresponds to a rigid-body rotation in Q-space that preserves the shape of the confidence ellipsoid. Thus, $R$ parameterizes a family of equivalent geometric solutions.
>
> Next, to select a suitable $R$, we impose the condition:
>
> - *"To ensure that the projection aligns with the total objective, the condition $u^T(CR)u \geq 0$ must hold. Otherwise, the scalarized lower bound could exceed the mean value, making it meaningless. In this sense, the rotation matrix $R$ serves as a generalized "clipping" operator, extending the conventional single-objective clipped double-Q learning (Fujimoto et al., 2018) to the multi-objective setting. How to compute $R$ is detailed in Appendix A."*
>
> That is, $R$ is chosen so that, after scalarization by $u$, the lower bound remains below the mean, mirroring the behaviour of clipped double-Q in the single-objective case. The geometric meaning of this "clipping" via $R$ is illustrated in the right panel of Figure 2.
>
> To summarize, **the goal of the process above is to derive a general form of the solution $A$ under the linear transformation.** Due to the page limitation, we could not explain the geometry meaning step-by-step (and not necessary w.r.t. solving the equation). In the revised version, we added some generic explanations to clarify the goal:
>
> - line 174-176: *"Now we construct a projected point on this ellipsoid considering $u$. A natural choice is applying a linear transformation:"*
>
> - line 178: *"we seek the general form of $A$"*
>
> &nbsp;
>
> **Q1.2 Similar holds for Eq. (10), where $C_{\text{clip}} = CR$is introduced to drop $R$, but never used again. One can assume that $C_{\text{clip}}$ became $L_{\text{clip}}$ through Sec. 4.2, but this connection can be made clearer, or avoid dropping $R$ altogether.**
>
> Indeed, $C$ can be any factorization of the covariance matrix. But, in Sec. 4.2, we specify that, when using Cholesky decomposition, $C$ becomes $L$ (we switch to the more standard notation for the Cholesky factor). Next, in Sec. 4.3, we specify that $L_{\text{clip}}$ is the clipped cholesky factorization.
>
> To make this clearer, we accepted the reviewer's suggestion and modified the expression as follows:
>
> - line 224-225: *"$L_{\text{clip}}$ is the clipped Cholesky factorization of the covariance matrix, i.e., $LR$"*
>
> We still find it useful to keep the notation $C_{\text{clip}}$ in Eq. (10), as it emphasizes that classical clipped double-Q can be seen as a special case.

---

> ### Author Response · Authors · 2025-11-22
> **Official responses of Submission24260 to Reviewer cZ7N - part 2**
>
> **Q2. About the assumption/restriction and the case of $C = -I$**
>
> No, there are no such restrictions.
>
> In our setting, $C$ is not an arbitrary symmetric matrix. As described in Sec. 4.2, $C$ is a square-root factor of a covariance matrix (positive-definite). **$C$ must have positive diagonal entries by definition.**
>
> The positive entries requirement is standard and conventional. For example, in the Cholesky factorization, although $LD$ is also a valid factor for any diagonal $D$ with $\pm 1$ entries, by rule, one always selects the factor with strictly positive diagonals to ensure uniqueness, continuity, and a well-defined algorithm (taking the principal square root at each step).
>
> So, $C = -I$ cannot occur. To avoid this confusion, we updated the sentence as follows:
>
> - line 179: *"If $\Sigma_Q$ is factorized into "squared-root forms" with positive entries:"*
>
> &nbsp;
>
> **Q3. About the meaning of prioritization and the role of $u$**
>
> The prioritization is defined by the ordering of the objectives in the Cholesky factorization, as introduced at the beginning of Sec. 4.2 and formalized in Eq. (11) via the chain rule.
>
> Cholesky decomposition is order-sensitive; putting x ahead of y or putting y ahead of x gives different $L$, corresponding to different decompositions of the joint distribution. Thus, prioritization means that **higher-priority objectives appear earlier in this chain rule**, so that lower-priority objectives are conditioned on them. This is exactly what we stated:
>
> - *"Specifically, each objective’s confidence bound is conditioned only on higher-priority objectives, with the highest-priority objective relying solely on its marginal distribution."*
>
> If the reviewer is asking **how** to determine the ordering, the prioritization itself is not explicitly given by the task, but essentially an inductive bias. It is defined by the user of the algorithm. As shown in the top row of Figure 4, if the improper ordering is used (unmatched inductive bias), then the performance degrades. **When safety is involved in the RL task, prioritizing safety in COP-Q is the best choice.**
>
> The scalarization vector $u$ plays a different role: it only specifies how the objectives are combined into a scalar performance measure, but it does not determine the priority order itself.
>
> &nbsp;
>
> **Q4. About the definition of uncertainty**
>
> In our manuscript, uncertainty refers to a quantitative measure of the potential **bias** in Q-value predictions.
>
> In off-policy RL with ensemble critics, this is typically captured by the disagreement between multiple independent Q-networks, e.g., SAC, TD3. In our multi-objective setting, where Q-values are vector-valued, we measure uncertainty by the **covariance matrix** of the Q-values.
>
> &nbsp;
>
> **Q5. About the level of optimism when facing uncertainty**
>
> Yes. The level of optimism when facing uncertainty is the same as in classical upper-confidence-bounds (UCB).
>
> In the single-objective case, if the Q-value has mean $\mu$ and standard deviation $\sigma$, an upper confidence bound takes the form $\mu + \beta \sigma$, where $\beta > 0$ is a hyperparameter. Intuitively, $\beta$ controls how far above the mean the agent is willing to look. A larger $\beta$ means the agent is more optimistic: it is more willing to trust high but uncertain estimates (taking more risk for potentially higher return). A smaller $\beta$ means the agent is more conservative.
>
> In our manuscript, this notion is generalized from the scalar case to the multi-objective setting. $\beta$ ($\chi^2$ here) still controls how "optimistic" the agent is with respect to the uncertainty in the (vector-valued) Q-estimates.
>
> We clarify this point by adding a sentence:
>
> - line 227-228: *"A higher $\chi^2_N(p_e)$ means being more optimistic, saying, take a higher risk to pursue a potentially higher return."*
>
> &nbsp;
>
> **Q6. About how to set the prioritization between objectives**
>
> Yes, how to set the prioritization does matter. In the original submission, we devoted a full paragraph in Sec. 5.1 (Objective ordering) to analyzing the impact of prioritization, with empirical results in the top row of Figure 4. There, we explicitly compare different objective orderings under the same scalarization and hyperparameters, and observe noticeable performance differences. In one sentence, **putting safety first is the best choice when safety is regarded as an objective in an RL task.**

---

> ### Author Response · Authors · 2025-11-22
> **Official responses of Submission24260 to Reviewer cZ7N - part 3**
>
> **Q7. About the minimal computational overhead**
>
> The "minimal computational overhead" has two concrete aspects in our approach:
>
> - **The number of critics:** For computing the covariance matrix of $N$ objectives, at least $N+1$ points are needed. (numerical stability issue and how to handle the reduced co-linear cases are given in Appendix B). We use this minimal number of critics.
>
> - **The efficiency of Cholesky decomposition:** For a symmetric, positive-definite matrix, Cholesky decomposition is known to be the most computationally efficient factorization method with $n^3/3$ complexity. LU, QR, SVD, etc., are all slower than it.
>
> To make this explicit, we add the following clarification:
>
> - line 252-253: *"This implementation introduces minimal computation overhead by using the minimal number of critics and the most efficient factorization."*
>
> &nbsp;
>
> Thank you again for the time and effort you put into reviewing our work. We hope these clarifications help convey the novelty and practical relevance of our approach more clearly. If you have any further concerns, we would be happy to discuss them.

---

### Official Review · Reviewer_URhw · 2025-10-29

**Soundness:** 3
**Presentation:** 3
**Contribution:** 3
**Rating:** 6
**Confidence:** 2

**Summary:**

COP-Q introduces a novel multi-objective Q-learning method leveraging Cholesky factorization to incorporate inter-objective covariance into uncertainty estimation. It prioritizes safety-critical objectives via ordered projection, yielding conservative TD targets for overestimation reduction and optimistic bounds for exploration. Evaluated on MuJoCo and SafetyVelocity-v1, COP-Q shows robust safety, competitive returns, and improved sample efficiency.

**Strengths:**

1. Innovative Uncertainty Modeling: First work to integrate Cholesky factorization into multi-objective Q-learning, capturing objective correlations and priorities (e.g., safety-first).

2. Strong Empirical Results: Outperforms baselines in safety-critical tasks while maintaining high returns.

3. Theoretical Soundness: Confidence bounds generalize clipped double-Q learning, with rigorous projection derivations.

**Weaknesses:**

1. Assumes Q-values follow multivariate Gaussian (Eq 5), but no ablation on its validity.

2. Limited Task Diversity: Experiments focus on locomotion; lacks MORL Pareto-frontier or high-dim task validation.

3. Variance in Exploration: REDQ+COP-OAC shows high variance in Humanoid (Fig 5), attributed to UTD ratio but unverified.

4. Sec 4.3 uses biased covariance estimator (denominator = N+1). Justify this choice.

5. Why there is no result about COP-Q-svc on halfcheetah experiment in Figure 4 top.

**Questions:**

Please refer to Weaknesses section.

---

> ### Author Response · Authors · 2025-11-22
> **Official responses of Submission24260 to Reviewer URhw - part 1**
>
> We appreciate the reviewer for giving these helpful comments. They are important to clarify the contribution of this work. Our detailed responses are given as follows:
>
> &nbsp;
>
> **Q1. Validity of multivariate Gaussian assumption**
>
> We use a multivariate Gaussian primarily for its **simplicity and analytical tractability**. It is an unbiased model, admits a clear geometric interpretation (via ellipsoid), and allows all key operations (e.g., projections, uncertainty measures) to be expressed in closed form. Gaussian assumptions are also quite common choices in uncertainty quantification. e.g. [1-3].
>
> In our paper, multi-objective uncertainty is measured by the covariance matrix of Q-values, so a Gaussian is a natural choice. While other priors (e.g., Dirichlet or Laplacian) are in principle possible, their multi-dimensional forms are more complex and would obscure the core contribution of this work. Exploring such alternatives is beyond the scope of this paper and left as interesting future work.
>
> In our original submission, the last section about limitations says:
>
> - *"First, the Gaussian assumption on Q-values (expectation of returns), although straightforward, might be oversimplified."*
>
> [1] *Nam, Daniel W., Younghoon Kim, and Chan Y. Park. "Gmac: A distributional perspective on actor-critic framework." International Conference on Machine Learning. PMLR, 2021.*
>
> [2] *Duan, Jingliang, et al. "Distributional soft actor-critic with three refinements." IEEE Transactions on Pattern Analysis and Machine Intelligence (2025).*
>
> [3] *Kendall, Alex, and Yarin Gal. "What uncertainties do we need in Bayesian deep learning for computer vision?." Advances in neural information processing systems 30 (2017).*

---

> ### Author Response · Authors · 2025-11-22
> **Official responses of Submission24260 to Reviewer URhw - part 2**
>
> **Q2. Limited task diversity and MORL benchmarks**
>
> Reviewer 6saY also raised a similar concern. Our response is as follows:
>
> In principle, COP-Q can indeed be extended to multi-objective RL. However, as the title indicates, this manuscript focuses on safety-critical RL tasks: (1) with an explicit safety objective or (2) with a safety cost constraint. **These settings differ fundamentally from standard MORL benchmarks, which makes a direct empirical comparison inappropriate:**
>
> - **First, the goals are different.** In safety-critical RL, the goal is to maximize return while ensuring safety, either by maximizing a safety reward or by keeping an unsafe cost below a threshold. In contrast, MORL benchmarks aim to approximate the entire Pareto front. The emphasis is on coverage and trade-offs across objectives, rather than on satisfying a safety requirement.
>
> - **Second, the main evaluation metrics are different.** Safety-critical RL typically evaluates return and safety. By contrast, the cited MORL works mainly report hypervolume as the key metric, measuring the quality of the approximated Pareto front. Thus, even if we were to run COP-Q on those benchmarks, the evaluation protocol and goals would not be aligned with the safety-centric questions we study here.
>
> - **Third, the episode termination and role of safety differ, especially in standard MuJoCo.** In standard MuJoCo control tasks, an episode terminates immediately when the robot falls. If we explicitly separate a "safety objective", the resulting Pareto front becomes essentially vertical to the safety axis: safety acts as a precondition for improving performance, rather than an objective to be smoothly traded off. This is precisely why almost **all MORL benchmarks do not treat safety as a separate objective.** For example, in MO-Gymnasium, none of the environments includes a safety objective.
>
> Although MORL and safety-critical RL are therefore not directly comparable in our current setting, we agree that studying COP-Q in MORL benchmarks is an interesting and important direction. In particular, it remains unknown how COP-Q would affect the hypervolume and the shape of the learned Pareto front. Would the impact be negligible, as in halfcheetah? Or would COP-Q primarily improve certain sub-regions of the Pareto front? There are still open questions.
>
> In our original submission, we have explicitly acknowledged this limitation and listed it as future work in the last section:
>
> - *"Third, our experiments did not evaluate the effectiveness of COP-Q in broader settings, such as MORL for approaching the Pareto front. This direction needs more investigation."*
>
> Moreover, to specify the scope of this study clearly in the early part of the paper, we adjust the research question as follows:
>
> - line 81-82: *"How can multi-objective uncertainty in Q-values guide exploitation and exploration in **safety-critical** deep Q-learning, thereby improving learning robustness, **safety compliance**, and sample efficiency?"*
>
> We also add the following sentence at the end of the related work section:
>
> - line 126: *"This paper fills this gap by explicitly incorporating full inter-objective covariance into Q-learning frameworks, particularly for safety-critical RL tasks."*
>
> In terms of high-dim task validation, as explained in our response to **[Q9]** of Reviewer 6saY: **Walker2d, ant, and humanoid are inherently challenging.** We agree that applying COP-Q to more complex tasks (e.g., autonomous driving) would further enhance the study, but it requires significant additional engineering. For this submission, we therefore keep it focused on the theory and methodology.

---

> ### Author Response · Authors · 2025-11-22
> **Official responses of Submission24260 to Reviewer URhw - part 3**
>
> **Q3. About the cause of high variances in exploration for humanoid**
>
> Thank you for bringing this issue to our attention. We did not attribute this to a high UTD ratio alone. In lines 417-420, we say,
>
> - *"The performance variance remains high for humanoid. Using more random seeds cannot narrow the variance. For some random seeds, the policy is always stuck in suboptimal points, probably due to the high UTD ratio of REDQ or simulation setups."*
>
> We actually carried out ablation studies specifically for humanoid. We tested the REDQ baseline (without active exploration) using the same 10 critics under different UTD ratios, ranging from 2 to 20. The results of the same 5 random seeds are listed in the table below. We see that both the mean and std of returns increase when UTD >= 8. However, the tendency is not perfectly monotonic, indicating random seed also plays a critical role.
>
> |return| UTD=2 | UTD=4 | UTD=8 | UTD=16 | UTD=20 |
> |------|-------|-------|-------|--------|--------|
> | mean |  2276 |  2364 |  5911 |   6546 |   6109 |
> | std  |  1023 |   996 |  5439 |   5202 |   5729 |
>
> Also, according to the original REDQ paper, this high variance does not occur for single-objective cases. So, probably, the high variance is a complex result of "multiple-objective + random seeds + high UTD + humanoid robot". Further experiments are needed to identify the true reason. We leave this for further work.
>
> We therefore modified our expression as follows in the updated version, acknowledging that the potential reason can be complex:
>
> - line 412-417: *"The variance of return remains high for humanoid. Using more random seeds cannot narrow the variance. For some random seeds, the policy is always stuck in suboptimal points. However, this high variance is not reported in the original single-objective REDQ (Chen et al., 2021). This high variance is likely due to the combination of high UTD and multi-objective critic learning. More investigations are needed to clarify the reason further."*
>
> &nbsp;
>
> **Q4. About the biased covariance estimator**
>
> We intentionally use a biased covariance estimator in Sec. 4.3, and this choice has been explained:
>
> - *Notably, when N = 1 (single-objective), equation 12 is exactly clipped double-Q learning (Fujimoto et al., 2018).*
>
> In other words, our primary design goal here is **compatibility with the classical clipped double-Q learning** when N = 1. Also, because we use only $N+1$ critics for $N$ objectives, using a biased estimator is a standard choice for such a small number of samples. This is not an ad-hoc choice: a similar biased variance estimate is also used in OAC [4] for constructing an optimistic upper bound of Q-values.
>
> [4] Ciosek, Kamil, et al. Better exploration with optimistic actor-critic. NeurIPS 2019.
>
> &nbsp;
>
> **Q5. About the training curves in Figure 4**
>
> Thank you for pointing out this issue. The results are not wrong, but our presentation makes it confusing. More details have been explained in our responses to **[Q1]** of the first reviewer 6saY. In short, **halfcheetah does not have a safety objective.** Therefore, **COP-Q-svc = COP-Q-vsc = COP-Q-vc.** It is a reference case without a clear prioritization between objectives.
>
> To avoid this confusion, we updated the top row of Figure 4, objective ordering. Now, halfcheetah and the other 4 robots use separate labels and legends. For halfcheetah, we explicitly mark the two curves as "COP-Q-cv" (orange curve) and "COP-Q-vc (default)" (red curve for consistency, ensuring that all default COP-Q use red).
>
> Additionally, to make readers aware of this key difference earlier, we add the following sentence when introducing MuJoCo environment for the first time at the beginning of Sec. 5:
>
> - line 267-268: *"Note that halfcheetah has no safety objective. It is selected as a reference case without clear prioritization between objectives."*
>
> And we also updated Table 1 by adding the abbreviations of objectives, making the differences of halfcheetah even clearer.
>
> &nbsp;
>
> We thank the reviewer again for the feedback. We hope that our responses address your concerns, and we would be happy to further discuss any remaining questions.

---

> > ### Comment · Reviewer_URhw · 2025-11-28
> >
> > I appreciate the authors’ detailed rebuttal, which has clarified many of my earlier concerns and improved my understanding of the paper. However, given the remaining limitations, such as the Gaussian assumption and the exploration of the Pareto front, I believe that my original score of 6 already represents a relatively positive assessment of the work. Therefore, I prefer to retain my initial evaluation.

---

> > > ### Author Response · Authors · 2025-11-28
> > >
> > > Thank you for your careful review, friendly score, and constructive feedback. They are valuable for further improving our study. Regarding the remaining concerns, we would like to address them briefly as follows.
> > >
> > > - **Validity of the Gaussian assumption:** Using whatever priors does *not* change the results of COP-Q, because the covariance matrix is computed from multiple points via a classical (frequentist) estimation. Only when using Bayesian estimation (e.g. NLL), assumed priors will make a difference. But we do agree that examining the Gaussian assumption can provide more insights about the biases.
> > >
> > > - **Exploration of the Pareto front:** We agree that examining the effectiveness of COP-Q in MORL w.r.t the Pareto front is important. We regard this as a valuable follow-up research topic. For this specific submission about safety-critical RL (as the title indicates), we prefer to keep it focused and not to include Pareto-front comparison due to its incompatibility with the safety requirement.
> > >
> > > Thank you again for your review and participation in the discussion.

---

### Official Review · Reviewer_6saY · 2025-10-30

**Soundness:** 3
**Presentation:** 2
**Contribution:** 3
**Rating:** 6
**Confidence:** 2

**Summary:**

This paper proposes Cholesky Ordered Projection Q-learning (COP-Q) to enhance safety-first exploitation and exploration for vector-valued Q-functions. In particular, COP-Q first introduce the generalized multi-objective confidence bounds for Q-values and then employ Cholesky factorization to encodes full multi-objective uncertainty following the priority structure of objectives. COP-Q achieves good performance and training efficiency compared with existing baselines on MuJoCo benchmarks.

**Strengths:**

1. This paper introduces Cholesky Ordered Projection into Multi-Objective RL so that different objectives can be considered with different priority.
2. COP-Q shows good performance and training efficiency compared with existing baselines on MuJoCo benchmarks.

**Weaknesses:**

### Concerns on Mujoco  benchmarks
1. In Figure 4, the training curve in halfcheetah seems to be wrong.
   1. There are only two curves in the figure.
   2. The blue curve (COP-Q-vsc) on top is exactly same as Cholesky on bottom
      1. Since Cholesky is same as COP-Q-svc in other environments, there is something wrong in the figure.

2. For baseline in MuJoCo, why doesn't COP-Q compare with MORL methods, such as RMORL [1], PGMORL [2], MORL-Adaptation [3], MO-MPO [4].

[1] He, X., Hao, J., Chen, X., Wang, J., Ji, X., & Lv, C. (2024). Robust multiobjective reinforcement learning considering environmental uncertainties. *IEEE Transactions on Neural Networks and Learning Systems*, *36*(4), 6368-6382.

[2] Xu, J., Tian, Y., Ma, P., Rus, D., Sueda, S., & Matusik, W. (2020, November). Prediction-guided multi-objective reinforcement learning for continuous robot control. In *International conference on machine learning* (pp. 10607-10616). PMLR.

[3] Yang, R., Sun, X., & Narasimhan, K. (2019). A generalized algorithm for multi-objective reinforcement learning and policy adaptation. *Advances in neural information processing systems*, *32*.

[4] Abdolmaleki, A., Huang, S., Hasenclever, L., Neunert, M., Song, F., Zambelli, M., ... & Riedmiller, M. (2020, November). A distributional view on multi-objective policy optimization. In *International conference on machine learning* (pp. 11-22). PMLR.

### Concerns on Constrained benchmarks

1. While this paper compares on SafetyVelocity-v1, it may have following concerns
   1. The feasible set of SafetyVelocity-v1 is not tight. The relationship between reward and cost are more likely to be independent to each other.
      1. For example, the agent can achieve the maximum rewards on a wide range of costs.
   2. This issue will simplify the testing cases.
2. Thus it is suggested to test on more benchmarks, whose feasible set is tighter.
   1. BulletSafetyGym:
      1. BallRun, BallCircle, DroneRun, DroneCircle, AntRun, AntCircle
   2. SafetyGymnasium:
      1. PointCircle1-v0, PointCircle2-v0, CarCircle1-v0, CarCircle2-v0,
3. It is suggested to compare with more recent methods, such as RLSF [1]
4. What are the criteria for selecting the cost threshold?
   1. It is better to test on multiple thresholds across a wide range of costs.
5. The cost in Figure 6 and Table E.3 seems to be unmatched.
6. It is better to list the results of PPOSimmerPID and CUP in Table E.3.

[1] Reddy Chirra, S., Varakantham, P., & Paruchuri, P. (2024). Safety through feedback in Constrained RL. *Advances in Neural Information Processing Systems*, *37*, 139938-139967.

### Other Concerns

1. In most figures, some methods seems that it hasn't converged within setting episodes.
   1. For example, walker2d and ant in Figure 5.
   2. For example, most on-policy method in Figure 6.
   3. What is the performance after convergence?
   4. Does COP-Q performs better than these methods after convergence?

**Questions:**

Please refer to the Weaknesses part.

---

> ### Author Response · Authors · 2025-11-22
> **Official responses of Submission24260 to Reviewer 6saY - part 1**
>
> We sincerely appreciate your careful review of our manuscript and your insightful, constructive comments. Our point-by-point responses are given below.
>
> **Q1. About the training curves in Figure 4.**
>
> Thank you for your careful review and for pointing out this issue. The results are not wrong, but we agree that our presentation makes it confusing.
>
> **For halfcheetah, COP-Q (default) = COP-Q-vc. There is no safety objective.**
>
> As shown in Table 1 (page 6), halfcheetah has no safety objective in the default setting. The two objectives are velocity (v) and control cost (c). We select halfcheetah as a reference case without clear prioritization between objectives. In COP-Q, the default ordering for halfcheetah is −vc. In our plotting code, we skip "s" when it does not exist. Consequently, the COP-Q-vsc curve and the default COP-Q-svc curve overlap in Figure 4, causing one of them to be invisible.
>
> To avoid this confusion, we updated the top row of Figure 4. Now, halfcheetah and the other 4 robots use **separate labels and legends**. For halfcheetah, we explicitly mark the two curves as "COP-Q-cv" (orange curve) and "COP-Q-vc (default)" (red curve for consistency, ensuring that all default COP-Q use red).
>
> Additionally, to make readers aware of this key difference earlier, we add the following sentence when introducing the MuJoCo environment at the beginning of Sec. 5:
>
> - line 267-268: *"Note that halfcheetah has no safety objective. It is selected as a reference case without clear prioritization between objectives."*
>
> We also updated Table 1 by adding the abbreviations of objectives, making the differences of halfcheetah even clearer.
>
> &nbsp;
>
> **Q2. About MORL baselines [1-4].**
>
> Thank you for providing these relevant MORL papers. In principle, our proposed COP-Q can indeed be extended to multi-objective RL. However, as the title indicates, this manuscript focuses on safety-critical RL: (1) with an explicit safety objective or (2) with a safety cost constraint. These settings differ fundamentally from standard MORL benchmarks, which makes a direct empirical comparison inappropriate:
>
> - **First, the goals are different.** In safety-critical RL, the goal is to maximize return while ensuring safety, either by maximizing a safety reward or by keeping an unsafe cost below a threshold. In contrast, MORL benchmarks aim to approximate the Pareto front. The emphasis is on coverage and trade-offs across objectives, rather than on satisfying a safety requirement.
>
> - **Second, the main evaluation metrics are different.** Safety-critical RL typically evaluates return and safety. By contrast, the cited MORL works mainly report hypervolume as the key metric. Thus, even if we were to run COP-Q on those benchmarks, the evaluation protocol and goals would not be aligned with the research questions we study here.
>
> - **Third, the episode termination and role of safety differ, especially in standard MuJoCo.** In standard MuJoCo, an episode terminates immediately when the robot falls. If we explicitly separate a "safety objective", the resulting Pareto front becomes essentially vertical to the safety axis: safety acts as a precondition=, rather than an objective to be smoothly traded off. This is precisely why almost **all MORL benchmarks do not treat safety as a separate objective.** For example, in MO-Gymnasium, none of the environments includes a safety objective.
>
> Although MORL and safety-critical RL are therefore not directly comparable in our current setting, we agree that studying COP-Q in MORL benchmarks is an interesting and important direction. It remains unknown how COP-Q would affect the hypervolume and the shape of the learned Pareto front. Would the impact be negligible, as in halfcheetah? Or would COP-Q primarily improve certain sub-regions of the Pareto front? These are still open questions.
>
> In our original submission, we have explicitly acknowledged this limitation and listed it as future work in the last section:
>
> - *"Third, our experiments did not evaluate the effectiveness of COP-Q in broader settings, such as MORL for approaching the Pareto front. This direction needs more investigation."*
>
> Moreover, to specify the scope of this study clearly in the early part of the paper, we adjust the research question as follows:
>
> - line 81-82: *"How can multi-objective uncertainty in Q-values guide exploitation and exploration in **safety-critical** deep Q-learning, thereby improving learning robustness, **safety compliance**, and sample efficiency?"*
>
> We also add the following sentence at the end of the related work section:
>
> - line 126: *"This paper fills this gap by explicitly
> incorporating full inter-objective covariance into Q-learning frameworks, particularly for safety-critical RL tasks."*

---

> ### Author Response · Authors · 2025-11-22
> **Official responses of Submission24260 to Reviewer 6saY - part 2**
>
> **Q3. The feasible set of SafetyVelocity-v1 is not tight. The relationship between reward and cost is more likely to be independent of each other. This issue will simplify the testing cases.**
>
> Thank you for this insightful comment. We address it in two parts: the (in)dependence between reward and cost, and the tightness of the feasible set.
>
> **Dependence between reward and cost in SafetyVelocity-v1**
>
> The reward and cost in SafetyVelocity-v1 are **not independent**. As explained in Appendix C, Eq. C.2, the reward is composed of a health term (precondition) + velocity term − control cost term (relatively smaller), while the safety cost is defined based on violations of the velocity constraint. Because the velocity term appears in both reward and cost, but with opposite signs (recall that in the primal-dual objective the cost enters as −$\lambda \times$ cost), increasing velocity will simultaneously **increase the reward** through the velocity component, but also **increase the cost** by violating the velocity constraint more frequently or severely. This shared dependence on velocity makes reward and (-cost) strongly and systematically negatively correlated, rather than independent.
>
> **Tightness of the feasible set**
>
> The tightness of the feasible set only impacts the difficulty of learning. The tightness itself is not a requirement for MORL or constrained safe RL. Therefore, we did not carefully check it when preparing the submission. Thank you again for pointing it out.
>
> In constrained RL, a "tight" feasible set means that the set of policies satisfying all safety constraints is small or restrictive: there is little room to improve the primary reward without quickly violating the constraint. This typically happens when **(1) safety requirements are strict:** the cost threshold is low, leaving very little budget for violations; and/or **(2) objectives are strongly negatively correlated:** actions that increase reward also tend to increase cost, creating a direct trade-off. SafetyVelocity-v1 falls into the second case, which is precisely what we refer to as a "tight" feasible set.
>
> To clarify these two aspects more explicitly, we added one sentence in Sec. 5:
>
> - line290-291: *"As the reward also contains the velocity term, the two objectives are systematically correlated, making the feasible set tight."*
>
> &nbsp;
>
> **Q4. Test on tighter BulletSafetyGym or SafetyGym benchmarks**
>
> Following our previous response to **[Q3]**, yes, we completely agree that these environments provide some benchmarks with even "tighter" feasible sets by adjusting the cost threshold. We indeed carried out experiments in the SafeNavigation benchmark in the Safety-Gymnasium. We selected 4 harder tasks: *SafetyPointPush1-v0*, *SafetyPointGoal2-v0*, *SafetyCarButton2-v0*, and *SafetyPointButton2-v0*. However, we found that the major bottleneck in SafeNavigation is not the correlated uncertainty, but the sparse cost signal makes the estimation bias in cost too large. **Consequently, COP-Q does not perform significantly better than other non-distributional RL baselines.** Addressing this issue requires extending COP-Q to distributional RL. We believe these concepts can be better addressed as two separate papers, as they focus on different challenges.
>
> We agree with the reviewer that we should make this point explicit to avoid cherry-picking. Therefore, we explained this in the discussion about limitations:
>
> - line 483-485: *"Fourth, COP-Q cannot compensate for the inherent high estimation biases when facing sparse cost signals, for example, in the SafeNavigation tasks. Extending COP-Q to distributional RL is a plausible future research direction."*

---

> ### Author Response · Authors · 2025-11-22
> **Official responses of Submission24260 to Reviewer 6saY - part 3**
>
> **Q5. Compare with RLSF**
>
> Thank you for refering to this relevant work.
>
> Generally speaking, our SafetyVelocity-v1 benchmark is primarily designed to evaluate **off-policy, Q-learning–based methods.** In constrained RL, the majority of existing methods are on-policy. Off-policy methods, although having higher sample efficiency than on-policy counterparts, are known to struggle with reliably satisfying cost constraints. In Figure 6, our goal is to show that COP-Q simultaneously achieves **(i) the high sample efficiency of off-policy methods** and **(ii) good cost-constraint satisfaction**, thereby addressing this known limitation. In our original submission, we explained in Sec. 5.2:
>
> - *"Compared to on-policy baselines, off-policy methods exhibit significantly higher sample efficiency. Among the off-policy methods, COP-Q shows robust safety performance, maintaining the cost below the threshold throughout the training."*
>
> From this perspective, adding yet another on-policy baseline, such as RLSF, would provide limited additional insight beyond what is already demonstrated.
>
> We agree that RLSF and other recent on-policy safe RL methods are important and relevant. To acknowledge this and guide future extensions, we have updated the baseline description section to read:
>
> - line 314-315: *"Other recent on-policy baselines, such as RLSF (Reddy Chitta et al., 2024), are also promising candidates for future comparison on this benchmark."*
>
> &nbsp;
>
> **Q6. About the cost threshold criteria**
>
> According to benchmarks like OmniSafe and SafetyGym, the cost threshold is typically set to around **half of the average velocity incurred by a conventional single-objective PPO policy within 1M env steps.** This choice is widely regarded as a reasonable value: the policy reaches the "unsafe" boundary frequently enough for updating the Lagrangian multiplier. If the threshold is set too high, the constraint is rarely active and requires many more env steps; if it is too low, the learning problem becomes trivial.
>
> To the best of our knowledge, very few constrained RL works sweep across different cost thresholds. For consistency and comparability with existing benchmarks, we adopt the same formulation in our experiments.
>
> &nbsp;
>
> **Q7. Unmatched results in Figure 6 and Table E.3**
>
> **Q8. Add PPOSimmerPID and CUP in Table E.3.**
>
> Thank you for your careful review. Indeed, they are unmatched. The two issues are: (1) the ordering of labels in the legend does not match, and (2) we rescale the cost by 0.1 in Table E.3.
>
> We have corrected the ordering of labels in Figure 6 and the cost in Table E.3 in the updated manuscript. Also, we added CUP in Table E.3. PPOSimmerPID has very close performance to PPOSaute because both of them are far from reaching the threshold, thus close to vanilla PPO. For clarity, we remove PPOSimmerPID from the baselines. As the results show, all on-policy methods perform poorly.
>
> We check-double-check again. They do match now. Thank you again.
>
> &nbsp;
>
> **Q9. About the performances after convergence**
>
> To be frank, for certain tasks, we cannot give a definitive performance after convergence for any of the reported methods, including COP-Q.
>
> For walker2d, ant, and humanoid, even SOTA methods using larger networks and complex techniques, such as TD7 [5], do not show flat learning curves after 5M environment steps. These tasks are inherently challenging: the robots have many degrees of freedom, and there is substantial potential to continually refine the gait and control strategy. As a result, the agent keeps exploring and improving, and the notion of "convergence" in the strict sense is often not reached within a reasonable training budget. This issue is even more pronounced for on-policy methods, which typically require much more environment interactions to stabilize because they cannot reuse experience via replay buffers.
>
> Therefore, in deep RL practice, **it is standard to evaluate algorithms in terms of sample efficiency:** given a fixed budget (e.g., 1M or 5M environment steps), which method achieves higher return and/or better safety?
>
> In this sense, our results show that, within the given training budget, COP-Q consistently achieves competitive or superior performance compared to the baselines. Beyond that regime, without prohibitively long runs, it is difficult to make rigorous statements about the exact asymptotic ranking of all methods.
>
> [5] *Fujimoto, Scott, et al. "For sale: State-action representation learning for deep reinforcement learning." Advances in neural information processing systems 36 (2023): 61573-61624.*
>
> &nbsp;
>
> We hope that the clarifications above help explain our evaluation choice and the practical limitations in claiming full convergence. If you have further questions or suggestions, we would be happy to discuss them.

---

> > ### Comment · Reviewer_6saY · 2025-11-26
> >
> > Thank you very much for the authors’ detailed responses. These replies have indeed addressed some of my concerns. Here are some comments for the rebuttal content.
> >
> > 1. About sparse signal:
> >    1. BulletSafetyGym seems to be dense signals.
> >       1. I list this benchmark first and Safety-Gymnasium is the second benchmark. I am a little confused as to why this first benchmark is not tested but the second benchmark is selected (This is not a question, no need to answer).
> >    2. We understand that Safety-Gymnasium contains sparse cost signal tasks. Thank you to the reviewer for adding it to the limitations.
> > 2. About sweeping across different cost thresholds:
> >    1. As a constrained RL, the constraint can be different in different cases.
> >    2. Although constrained RL doesn't need to sweep across different thresholds without retraining, it should have the ability to deal with different constraint thresholds after retraining.
> >    3. For example, in this paper, the threshold is $L_1$.
> >       1. Does COP-Q only perform well for $L_1$?
> >       2. If the threshold is $L_2$, can COP-Q be trained under this constraint and perform well?
> > 3. About convergence:
> >    1. I agree RL algorithm should consider both final performance and training efficiency.
> >    2. I point out this point because the performances of some baselines still increase rather than oscillate.
> >    3. I am just curious where these methods will eventually stop growing.
> >
> > I have given a very friendly initial score, combined with the limitations mentioned in the rebuttal, and I believe that this article has not yet reached the level of acceptance without doubt. Taking all the responses into account, I am inclined to keep my score.

---

> > > ### Author Response · Authors · 2025-11-26
> > >
> > > Thank you for your timely response and follow-up questions. We would like to clarify the remaining points as follows.
> > >
> > > 1. Sparse signals and BulletSafetyGym
> > >
> > > - We tested SafetyGym when preparing the initial submission, not during the rebuttal. Perhaps this missing point explained why the reviewer was confused about our choice.
> > >
> > > - We inspected BulletSafetyGym. Its tasks (circle, gather, reach, run) are highly similar to SafetyGym, and except for run (which is also velocity-constrained), the costs in BulletSafetyGym are indeed *sparse*, as explicitly stated on their GitHub page. We chose SafetyGym because it was already integrated into OmniSafe [1], which makes grid experiments much easier to conduct. And we finally found that, for sparse costs, the accuracy of critic learning is the fundamental bottleneck. That makes all point value-based methods perform similarly.
> > >
> > > 2. Sweeping across different cost thresholds
> > >
> > > We agree that evaluating across multiple cost thresholds is important, and we will add such ablations in the next version of the paper. However, retraining all baselines for several thresholds is time-consuming, so we are unsure whether we can complete this during the rebuttal. We will keep you updated.
> > >
> > > 3. Convergence
> > >
> > > We understand the concern. Indeed, if we continued training, some baselines’ curves would keep improving. 1M–3M (sometimes 5M) env steps are common for MuJoCo. If we want to compare the converged performance within this window, a practical way to accelerate is to increase network capacity or adopt distributional RL methods such as TQC [2]. We see this as a valuable direction in future work.
> > >
> > > Thank you again for your careful review, constructive suggestions, and friendly score. If you still have any follow-up questions, we would be happy to continue the discussion.
> > >
> > > [1] *Ji, Jiaming, et al. "Omnisafe: An infrastructure for accelerating safe reinforcement learning research." Journal of Machine Learning Research 25.285 (2024): 1-6.*
> > >
> > > [2] *Kuznetsov, Arsenii, et al. "Controlling overestimation bias with truncated mixture of continuous distributional quantile critics." International conference on machine learning. PMLR, 2020.*

---

### Author Response · Authors · 2025-11-22
**General comment for the 1st-round discussion**

Dear all,

The authors are grateful for the effort the reviewers put into reviewing our paper and giving insightful, constructive feedback, which has led to improvements in our manuscript.

We have uploaded a revised manuscript with all changes marked in blue text, titles, or boxes. Please refer to our point-by-point responses for a detailed guide to the revisions. We hope this updated version can facilitate the discussion between the authors and reviewers.

---

### Author Response · Authors · 2025-12-03
**Brief Factual Summary of Rebuttal**

We sincerely thank all reviewers for their reviews and discussions. To facilitate the evaluation of our study, we hereby provide a concise **factual summary** of the rebuttal for AC and all readers.

&nbsp;

**Main reviewer concerns**

**1. Theoretical clarity:** (1) Gaussian assumption for vectorized Q-values. (2) The choice of a biased covariance estimator. (3) Definitions of some key concepts like "prioritization", "uncertainty", and "optimism level". (4) Some math notations are confusing. (5) The missed underlying assumption for the rotation matrix computation.

**2. Experimental design:** (1) Confusing overlap in halfcheetah curves. (2) Lack of MORL / recent safe RL baselines. (3) Fixed cost thresholds (no sweeps) for constrained RL tasks. (4) The solution for the selected constrained RL tasks seems not tight enough. (5) Incomplete convergence in training. (6) Inconsistent results between figures and tables in the appendix.

&nbsp;

**Main clarifications and revisions:**

**1. Concepts and assumptions**

- Clarified that the Gaussian assumption was used for simplicity and traceability.

- Explained that a biased covariance estimator was chosen on purpose to ensure that the classical clipped double-Q learning is a reduced special case of our COP-Q.

- Clarified the key definitions of prioritization, uncertainty, and optimism level in the updated manuscript.

- Modified the math notations throughout the paper for consistency.

- Explained that there are no restrictions on the rotation matrix in theory.


**2. Experimental designs**

- Fixed the confusing training curves for halfcheetah, updated the corresponding Fig. 2.

- Explained that MORL is not the focus of this paper on safety-critical RL; Acknowledged MORL as future work. Explained that adding new on-policy baselines does not bring added value for this work, focusing on off-policy approaches.

- Argued that most existing works do not sweep over the cost threshold in constrained RL; Acknowledged this point as future work.

- Explained that the solution for SafetyVelocity is tight. For the recommended new safe RL tasks, we explained that their sparse costs make COP-Q difficult to apply directly.

- Explained that comparing the sample efficiency is common in RL, and for some challenging tasks (e.g., humanoid), reaching convergence is time-demanding.

- Fixed the inconsistent results caused by the wrong model name order in the legend box.

&nbsp;

**Scores:**

- Two reviewers with an initial score of 6 explicitly indicated that they prefer keeping the initial scores.

- The other two reviewers explicitly indicated that they would increase their scores.

&nbsp;

We hope that this factual summary is helpful for a better understanding of our work and the rebuttal process.

---

### Meta-Review · Area_Chair_ZyGp · 2025-12-24

**Summary:**

The authors noted that in vector-valued Q-learning with correlated objectives, naive independent or scalarized uncertainty estimates may destabilize learning and lead to unsafe behavior. To resolve this issue, the authors propose an interesting Cholesky Ordered Projection Q-learning (COP-Q) method. Experimental validation of the proposed methods has been done on MuJoCo and SafetyVelocity-v1.

The overall rating of the paper is paper is mixed. During the rebuttal, the authors have partially resolved the concerns from the reviewer, while still having a few issues unclarified. These issues include: (1) for constrained RL, how the method perform under different constraint thresholds has yet been tested; (2) the experiment does not show stabilized performance of the compared benchmarks, leading to a question whether COP-Q still outperforms the other algorithms after convergence, or it just starts improving earlier while performs less well after convergence; (3) the Gaussian assumption is not justified; (4) the inability to deal with environments with sparse reward signals.

For the other limitations, the AC thinks the authors have clearly addressed them, and the reviewers agree to rise their score. The one reviewer who gave a score of 2 has not responded to the authors. According to the AC's reading, we think the score will also rise a little bit if the reviewer had the chance to make adjustment.

Overall, the AC thinks the paper is a marginal paper. And the AC has to regretfully decide a reject for the paper.

**Reviewer Concerns:**

Addressed:
(2) cZ7N: Several technical and presentation questions.
(3) xEpw: All concerns from the reviewer (confirmed by reviewer's comment)

Not Addressed:
(1) 6saY: Lack of ability to deal with sparse reward signals. (The authors provide some justification, yet not fully address this issue)
(2) 6saY: The comparison is made before the compared benchmarks converge, and the experiment under different constraint thresholds are not done. (The authors provide some justification, yet not fully satisfactory)
(3) URhw: Assumes Q-values follow multivariate Gaussian (Eq 5), but no ablation on its validity. (The authors justify by citing a few other papers, yet the reviewer is asking for ablation evidence and wants to know whether it is really reasonable of just a technical artifact).

**Reviewer Scores:**

Though the score of xEpw has not changed on system, she/he mentioned the increase of score from 4 to 6. The score from reviewer cZ7N will likely increase from 2 to 4 or 5. However, the overall rating of paper is still marginally below the conference's threshold.

---

### Decision · Program_Chairs · 2026-01-26

Reject